# Spiral phyllotaxis predicts left-right asymmetric growth and style deflection in mirror-image flowers of *Cyanella alba*

Caroline Robertson [1,7], Haoran Xue [2,7], Marco Saltini [3,7], Alice L. M. Fairnie [4], Dirk Lang[5], Merijn H. L. Kerstens [6], Viola Willemsen[6], Robert A. Ingle [1], Spencer C. H. Barrett [4], Eva E. Deinum [3], Nicola Illing [1] & Michael Lenhard [2] ✉

Many animals and plants show left-right (LR) asymmetry. The LR asymmetry of mirror-image flowers has clear functional significance, with the reciprocal placement of male and female organs in left- versus right-handed flowers promoting cross-pollination. Here, we study how handedness of mirror-image flowers is determined and elaborated during development in the South African geophyte *Cyanella alba*. Inflorescences of *C. alba* produce flowers with a largely consistent handedness. However, this handedness has no simple genetic basis and individual plants can switch their predominant handedness between years. Rather, it is the direction of the phyllotactic spiral that predicts floral handedness. Style deflection is driven by increased cell expansion in the adaxial carpel facing the next oldest flower compared to the other adaxial carpel. The more expanding carpel shows transcriptional signatures of increased auxin signaling and auxin application can reverse the orientation of style deflection. We propose that a recently described inherent LR auxin asymmetry in the initiating organs of spiral phyllotaxis determines handedness in *C. alba*, creating a stable yet non-genetic floral polymorphism. This mechanism links chirality across different levels of plant development and exploits a developmental constraint in a core patterning process to produce morphological variation of ecological relevance.

Left-right (LR) asymmetry is a fascinating feature of many plants and animals. Striking examples include the asymmetric placement of internal organs in vertebrates and the left versus right coiling of snail shells[1,2]. Such LR asymmetries raise three fundamental questions. First, how is symmetry broken in a consistent manner to determine left and right? Second, how is this translated, from genes to cellular and tissue level processes, to produce asymmetric morphologies? Third, what is the biological function of LR asymmetry? The first two questions are being answered for the LR asymmetry of internal organs in vertebrates and shell coiling direction in snails[3–8]. In both systems the inherent chirality of different cytoskeletal elements appears to be exploited for the initial symmetry breaking event, which is then translated into an

[1]University of Cape Town, Department of Molecular and Cell Biology, Rondebosch 7701, South Africa. [2]University of Potsdam, Institute for Biochemistry and Biology, Karl-Liebknecht-Str. 24-25, D-14476 Potsdam-Golm, Germany. [3]Mathematical and Statistical Methods (Biometris), Plant Science Group, 6708 PB Wageningen, The Netherlands. [4]Department of Ecology and Evolutionary Biology, University of Toronto, 25 Willcocks Street, Toronto, Ontario M5S 3B2, Canada. [5]University of Cape Town, Department of Human Biology, Observatory 7925, South Africa. [6]Laboratory of Cell and Developmental Biology, Wageningen University, Droevendaalsesteeg 1, 6708 PB Wageningen, the Netherlands. [7]These authors contributed equally: Caroline Robertson, Haoran Xue, and Marco Saltini. ✉e-mail: michael.lenhard@uni-potsdam.de

asymmetric morphology via a partially conserved pathway. In mouse embryos, inherently chiral motile cilia of cells in the node cause a left-ward flow of extracellular fluid that breaks symmetry[5,6,9]. In *Lymnaea* snails a genetic polymorphism determines coiling direction, with the dominant *D* allele, which encodes a nucleator for actin filaments, causing right-handed coiling[3,7,10]. In both cases, LR-asymmetric upre-gulation of *Nodal* and *Pitx2* expression translates the initial symmetry breaking event into an asymmetric morphology. Despite this progress in understanding the molecular basis of handedness, the functional relevance of LR asymmetry in these examples remains largely unclear.

Several examples of LR asymmetry and left- or right-handed helical growth also occur in plants[2]. These include the left-right asymmetry of leaves in several species[11], left- or right-handed helical arrangement of leaves and flowers around the stem (i.e. phyllotaxis)[12], the twining of tendrils[13], and the left- or right-handed helical growth of mutants[14]. Several important components that underlie helical growth per se have been identified, such as a polar auxin transport-based mechanism that determines the initiation of organs at the shoot meristem in a phyllotactic spiral and indirectly seems to cause LR asymmetries in leaves[15,16], or gelatinous G-fibres in twisting tendrils[17]. But how the handedness of the resulting helices is determined remains unknown. The important exception are mutants with consistently left- or right-handed helical growth[14]. Most of these result from mutations in tubulin or microtubule-associated proteins that cause a spiral arrangement of cortical microtubules, with the handedness of this spiral determining the handedness of helical growth at the cell or organ level[18,19]. These mutants underscore the value of genetic poly-morphisms causing helical growth with a consistent orientation for identifying the molecular determinants of handedness.

Mirror-image flowers (enantiostyly) represent a case of LR asym-metry in plants for which a clear functional relevance has been demonstrated and where - at least in some taxa - the handedness of the flowers is genetically determined[20–22]. This provides an opportunity to identify both the molecular and developmental control of handedness and its adaptive significance[23]. In enantiostylous flowers the style is deflected to the left- or right-side of the midline that runs along the dorsoventral axis of the flower[24], and in most species a pollinating anther is deflected in the opposite direction (Fig. 1). Enantiostylous species are classified based on the distribution of left- and right-styled flowers on individuals[24]. In monomorphic enantiostyly, both forms of flower can be found on a single individual, whereas in dimorphic enantiostyly a plant only produces left- or right-styled flowers, called L-morph or R-morph plants, respectively.

Enantiostyly is reliably reported from 11 unrelated angiosperm families and represents a striking case of convergent evolution in form and function. Most enantiostylous species are of the monomorphic type[23,24], with dimorphic enantiostyly reliably reported from just two monocotyledonous lineages: Pontederiaceae (*Heteranthera, Mono-choria*) and Haemodoraceae (*Wachendorfia, Barberetta*). A third instance in Tecophilaeaceae (*Cyanella*) has been reported[20], but this has not been fully confirmed. Experimental studies have demonstrated that enantiostyly represents a floral adaptation that promotes cross-pollination and reduces sexual interference between male and female function in hermaphroditic flowers[25,26]. It does so by optimizing the placement of pollen on the bodies and wings of pollinating insects for efficient disassortative pollen transfer to styles on the opposite morph. Studies using genetic markers indicate that dimorphic enantiostyly promotes significantly higher levels of outcrossing than the mono-morphic condition[21,27]. The efficiency of dimorphic enantiostyly in promoting disassortative pollination has also been demonstrated using fluorescently labelled pollen grains in *Wachendorfia paniculata*[28]. Similar studies in the sister genus, *Barberetta aurea* reported that transfer of fluorescent dye occurred mainly between reciprocally positioned floral organs[29]. Thus, at least for some species/pollinator combinations the pollen that left-styled (L) flowers deposit

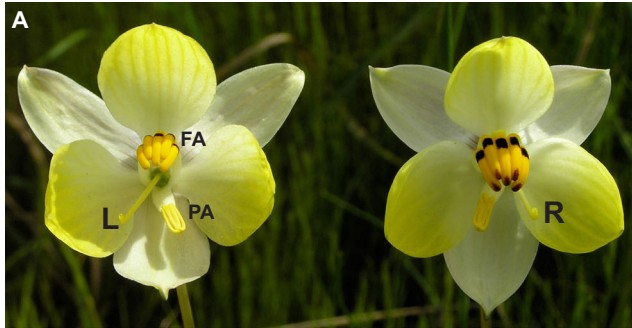

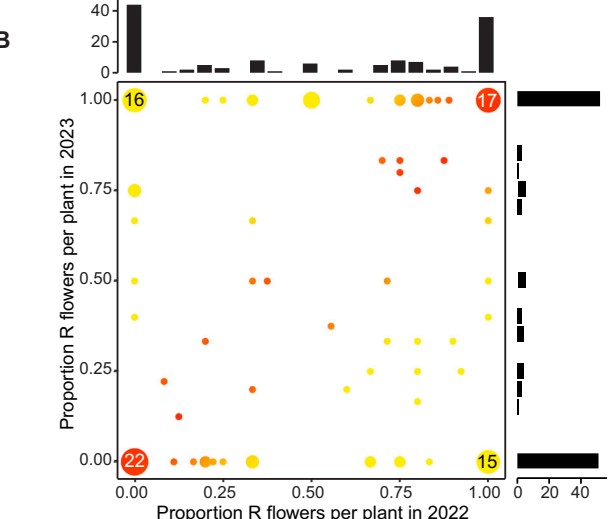

**Fig. 1 | Floral handedness has no simple genetic basis in *C. alba*. A** Image of L- and R-flowers of *Cyanella alba* subsp. *flavescens*. Feeding anthers (FA) and polli-nating anther (PA) are indicated. L and R indicate left- and right-deflected styles. **B** Distribution of floral phenotypes of the same cohort of 135 plants in 2022 and 2023. The colour gradient indicates the consistency in floral orientation between years (red: fully consistent, yellow: fully inconsistent). Black bars on the outside show marginal distributions.

on the right side of visiting insects is transferred efficiently to the stigmas of right-styled (R) flowers, and vice versa. The mode of inheritance of dimorphic enantiostyly has only been determined in *Heteranthera multiflora* (now *H. missouriensis*), where handedness is determined by a genetic polymorphism segregating in a Mendelian manner[22]. Plants that carry the dominant allele at the polymorphic locus form R-flowers, whereas the homozygous recessive genotype produces L-flowers.

Here, we investigate the molecular and development basis of enantiostyly in *Cyanella alba* subsp. *flavescens* (hereafter *C. alba*), a long-lived, deciduous geophyte restricted to the Western and North-ern Cape regions of South Africa[30]. Plants possess deep-seated corms that produce a basal rosette of narrow erect leaves after winter rains, with flowering occurring from early August to early October. The inflorescence is a compact raceme with very little stem elongation, and flowers are borne individually on long pedicels (10–20 cm), giving the appearance of solitary flowers emerging from the rosette. Each plant produces a small number (generally not more than 15) of long-lived flowers during the flowering season, but daily display size is most commonly one to two flowers[31]. *C. alba* is a candidate for showing dimorphic enantiostyly based on previous work[20,24,32], as the majority of plants were reported to form flowers of only one handedness and only a few plants produced both types of flowers[32]. Thus, floral hand-edness in *C. alba* may have a genetic basis.

**Table 1 | Numbers of SNPs associated with left- and right-styled phenotypes in *Cyanella alba* subsp. *flavescens***

| | SNPs in which L-plants are heterozygous | SNPs in which R-plants are heterozygous |
|---|---|---|
| Illumina2021 dataset (25 individuals) | 10 | 12 |
| Illumina2022 dataset (20 individuals) | 12 | 17 |
| Combined dataset (45 individuals) | 0 | 1 |

The number of SNPs in each dataset where 80% or more individuals of a given handedness are heterozygous at the SNP site is given. The analysis was restricted to the protein-coding regions of the *C. alba flavescens* genome.

Our investigation used a sequencing-based approach to search for the presumed genetic basis of floral handedness in *C. alba*. By analogy to other similar floral polymorphisms and to enantiostyly in *H. missouriensis*, we hypothesized that floral handedness in *C. alba* was under the control of a single *ENANTIOSTYLY* (*E*) locus with a dominant and recessive allele. Under such a scenario, disassortative (intermorph) mating between L- and R-morphs would produce an equal ratio of L- and R-morphs in the progeny, with one morph homozygous recessive and the other heterozygous at the *E* locus. Our study aimed to answer three questions. First, is there indeed a genetic polymorphism that determines floral handedness in *C. alba*? Second, what is the symmetry-breaking event that determines the direction of style deflection in flowers? Third, which cellular and developmental processes cause consistent style deflection to one side?

## Results

### Sequencing-based analysis and transcriptomics do not support a simple genetic polymorphism for floral handedness

To determine whether there is a genetic polymorphism determining handedness, we established a reference genome assembly from an individual that had formed seven consistently R-flowers in 2021 (see Supporting Information Text). We performed Illumina whole-genome sequencing on 25 *C. alba* individuals that had four or more flowers of consistent handedness in 2021 (15 L and 10 R), and 20 with six or more flowers of consistent handedness in 2022 (11 L and 9 R; see Supporting Information Text). In addition, we performed RNA-sequencing of dissected styles from bud stages covering the critical time window when the orientation of style deflection is determined (see Supporting Information Text). Using this data, we asked whether: (i) there is a bi-allelic locus associated with style orientation; (ii) there is a hemizygous region determining handedness, as demonstrated in several heterostylous species[33–35]; (iii) whether there are any transcripts expressed exclusively in plants with one style orientation. To answer these questions, we performed: (i) a genome-wide association study (Table 1) (ii) searched for a genomic region exclusively present in plants of one style orientation using both coverage- and *k*-mer based approaches (Fig. S1, Table S1), and (iii) identified differentially expressed genes between plants with the different style orientations (Supplementary Data 1). However, as detailed in the Supporting Information Text, none of these approaches provided any convincing evidence for a simple genetic polymorphism underlying floral handedness in *C. alba*. For example, no orientation-associated single-nucleotide polymorphisms (SNPs) were shared between the datasets from the two years and the combined dataset (Table 1), and the *k*-mer based approach identified more right-associated *k*-mers in the 2021 dataset, but many more left-associated ones in the 2022 dataset (Fig. S1).

### Switching floral handedness on individual plants between flowering seasons indicates that the direction of style deflection is not genetically determined

The above conclusion that there is unlikely to be a genetic basis to handedness is further supported from our demographic studies of individually marked plants during the flowering seasons of 2022 and 2023. In 2022, 188 plants in three populations were permanently marked with metal pins, and in 2023 we recovered 135 of these plants. We followed the plants throughout the duration of both flowering seasons and recorded the number of L- and R-flowers produced by each individual (Fig. 1B). This confirmed that flower orientation is not random, as the majority of plants produced only flowers of one orientation (59% and 76% of plants in 2022 and 2023, respectively). By contrast, a random process would result in a unimodal distribution of per-plant proportions of R-flowers centered around 50%. Strikingly, of the 44 plants recovered in 2023 that had formed exclusively L-flowers in 2022, 22 retained their handedness and 22 switched handedness in 2023. Of the latter, 16 produced exclusively R-flowers and 6 produced both L- and R- flowers (proportion of R-flowers within a plant ranged from 40-75%). Conversely, of the 36 plants recovered in 2023 that had exclusively R-flowers in 2022, 19 switched their handedness in 2023. Again, the same trend was seen as above with the majority, 15 plants, producing exclusively L-flowers in 2023, and the remaining four plants (21%) producing both flower types (proportion of L-flowers within a plant ranged from 33 to 60%). Thus, these results demonstrate that the handedness of flowers produced by an individual plant during one flowering season is highly consistent, but that this consistent handedness can switch within an individual from one year to the next. This finding supports the above conclusion that floral handedness in *C. alba* is not determined by a genetic polymorphism and suggests that the species exhibits monomorphic enantiostyly and not dimorphic enantiostyly as earlier assumed.

### The handedness of the phyllotactic spiral predicts the consistent bias in floral handedness within plants

The lack of direct genetic control of handedness raises an intriguing question: How can flowers in an inflorescence consistently develop L or R style deflection without genetic input, and how can they switch their annually consistent bias for one side to the other in the next year? One developmental process with a highly consistent handedness over the growing period of an inflorescence is the direction of the phyllotactic spiral. Its handedness is also known to cause LR asymmetries in leaves[15,16]. *C. alba* shows spiral phyllotaxis (Fig. S2), with organs initiated at the shoot apical meristem in either a clockwise or counter-clockwise direction. Changes in the handedness of the phyllotactic spiral produced by one shoot meristem are exceedingly rare[36], providing stability in any one year. This also holds true in *C. alba*, as none of 133 plants we sampled showed a consistent handedness reversal of the phyllotactic spiral. Moreover, studies of many angiosperm species indicate that the handedness of the phyllotactic spiral is not under genetic control, and that initiating shoot meristems randomly establish either a clockwise or counter-clockwise spiral with equal probability[12]. This pattern is also true for the individual lateral meristems on a single plant. Thus, it is likely that the shoot meristems initiated in successive years by *C. alba* randomly adopt a clockwise or counter-clockwise orientation. We therefore asked whether the direction of style deflection in *C. alba* flowers is predicted by the handedness of the phyllotactic spiral.

To address this question, we determined the handedness of the phyllotactic spiral and the orientation of the styles on 132 *C. alba* plants in the 2023 flowering season. Style orientation was determined from between one and seven open flowers and sufficiently mature buds per plant (average 2.74, median 3 flowers per plant; a small number of flowers with unclear style orientation were excluded). This revealed a strong correlation between the direction of the phyllotactic spiral and

floral handedness, with a correlation coefficient of 0.82 (0.76–0.87). Most plants with a counter-clockwise or right-handed phyllotactic spiral (defined from older to younger organs) formed only R-flowers, and vice versa for plants with a clockwise or left-handed phyllotactic spiral. Twenty plants produced both R- and L-flowers, and there were very few plants (5/132) where floral handedness consistently contradicted that predicted from the phyllotaxis direction. There were 16 mixed plants for which we could score more than two flowers each; in twelve of these plants, the majority of flowers conformed to the handedness predicted by phyllotaxis. Very similar results were observed in 53 plants analyzed in 2024 (Fig. S2J). Thus, the handedness of the phyllotactic spiral in *C. alba* generally predicts floral handedness, suggesting that it provides the consistently biased input into flower development to ensure the non-random distribution of style deflection of individual plants through time.

We next asked whether deviations from the floral handedness predicted by phyllotaxis were randomly distributed in an inflorescence. For this we plotted the fraction of such deviations per absolute flower position in an inflorescence and compared it to the overall average of all flowers (Fig. 2B). The distribution of deviations was non-random across the different positions ($p < 0.0001$, $X2$-test), with higher deviation rates at the very first and at higher positions beyond position 5, and the lowest deviation rates at positions 2 to 4, i.e., the very first and later flowers during the season were more likely to have deviant handedness. The reason for this non-random distribution of errors is unclear at present, but it may suggest an additional flower-flower interaction that cannot operate on the first flower and becomes weaker over time.

## Style deflection results from differential elongation of the two adaxial carpels

The styles of *C. alba* are initially straight with no discernible difference between L- and R- flowers (Fig. 3A). When styles of early buds reach 70% or more of the length of the pollinating anther, they begin to deflect slightly to either the left or right of the midline. The stigma tip can also be slanted at this stage, with the slant facing the direction to which the style will be deflected. In mid-stage buds, one to two days before anthesis, the style is clearly deflected from the midline, and in open flowers the style and pollinating anther are fully deflected to opposite sides of the flower. The style is relatively straight, with the exception of the stigma tip curving sharply upwards. The style movement to one side can be observed using time-lapse microscopy in dissected flowers cultivated in vitro (Video S1). The hinge-like motion of the style is coupled with expansion and elongation of the ovary, as confirmed by measuring the distances between three marked spots on the ovary, which increased by 14% and 10% in the length and width direction over the course of the video, respectively (Video S1). These observations suggested that style deflection is driven by differential elongation of the carpels. The *C. alba* gynoecium consists of three carpels, with the two adaxial carpels lying on either side of the dorsoventral midline of the flower and the third carpel below, such that the midline bisects it (Fig. 3B). If one of the two adaxial carpels elongated more than the other, this could push the style away from the more strongly elongating carpel.

To test the feasibility of this mechanism and investigate the contribution of differential elongation of different carpel walls, we developed a biophysical model of the *C. alba* pistil. Our aim was to test whether differential elongation of different carpels can drive style deflection to the extent observed in *C. alba* ovaries. As individual ovaries varied substantially in size and shape, we opted for a model that captures organ shape in a simple description, to which we could fit the individual ovaries, rather than a model reproducing realistic organ shapes. To this end, we developed a bead-spring model with as few springs per carpel as possible to capture the essence of the developmental process[37]. Bead-spring mechanical models are well-suited for

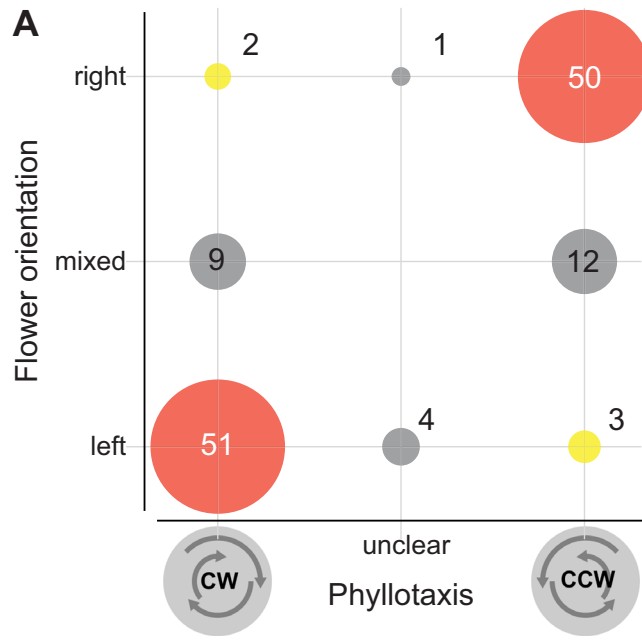

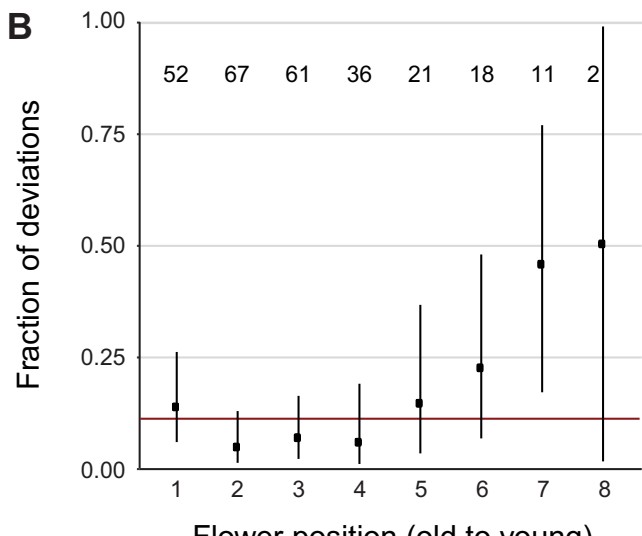

**Fig. 2 | Direction of the phyllotactic spiral determines floral handedness in *Cyanella alba* subsp. *flavescens*. A** Correlation of phyllotactic direction (CW: clockwise, CCW: counter-clockwise) and floral handedness. Numbers in the diagram indicate the number of individual plants. 'Unclear' phyllotaxis indicates plants with individual divergence angles that are inconsistent with their overall phyllotaxis orientation. See Materials and Methods in Supporting Information for more details and our interpretation of these cases. **B** Distribution of deviations (flowers with the opposite style deflection compared to the orientation predicted based on phyllotaxis) across flower positions in an inflorescence. Numbers in diagram indicate the number of flowers scored at each position. Horizontal red line indicates average fraction of such deviations over all flower positions. Dots are observed values and vertical lines indicate 95% confidence intervals for the fraction of deviations per position calculated based on a binomial test.

identifying differential growth patterns without the need for complex geometries or full organ-level modelling[37]. We used a three-dimensional polyhedral model to represent the ovary, which is composed of a triangular prism topped by a truncated pyramid (Fig. 3C and Supporting Information Text; Figs. S3, S4; Tables S2–S8). Deflection of the truncated face of the pyramid such that it is no longer parallel to the base of the triangular prism indicates style deflection. We applied this model to images of six ovaries in two steps. First, we identified for

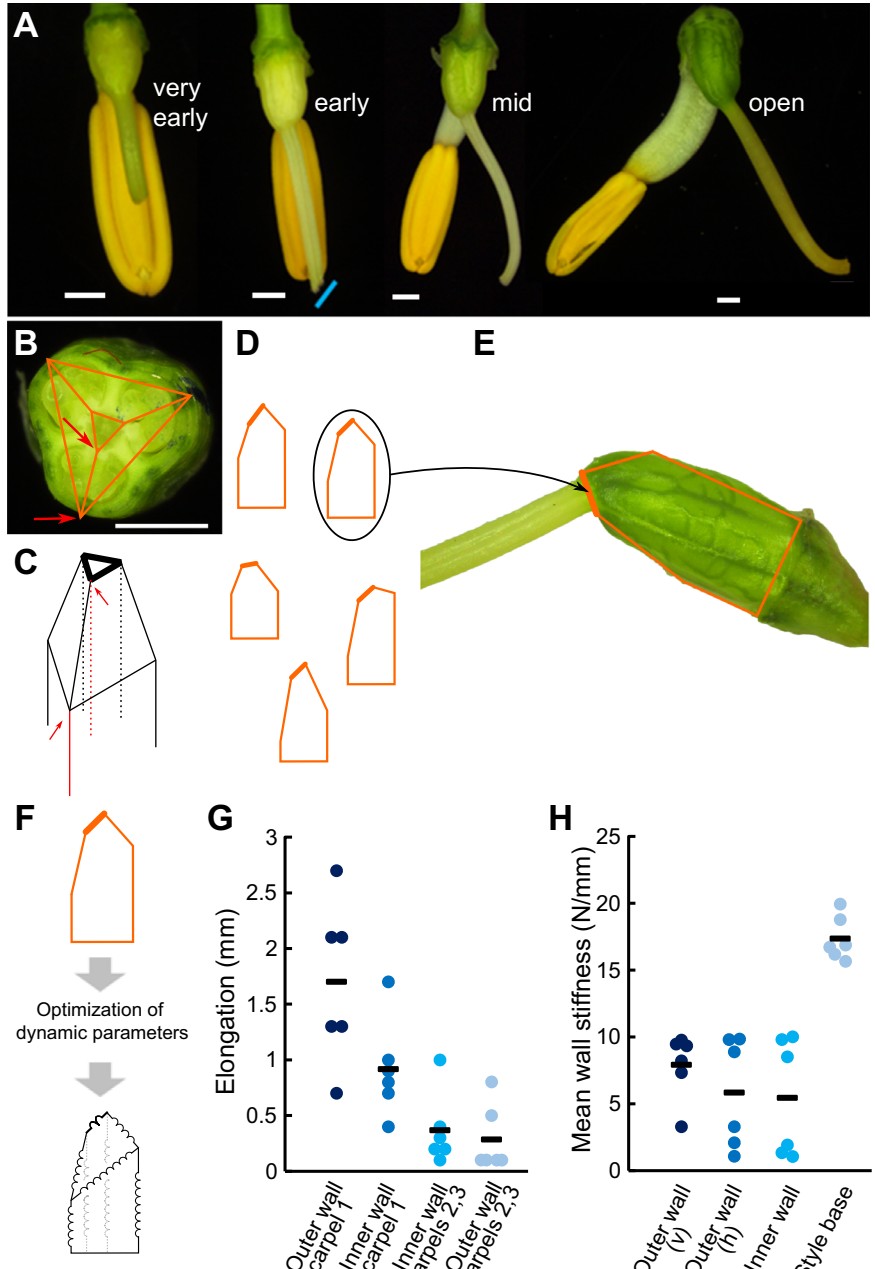

**Fig. 3 | Schematic of the data-driven and mechanistic models and analysis of ovaries of *Cyanella alba* subsp. *flavescens*. A** Stereomicroscope images of *C. alba* buds at four stages of development reveal the developmental sequence of stylar bending. Blue line indicates the slant of the stigma in early-stage buds. Scale bars are 1 mm. These dissections were done twice independently in 2021 and 2022 with very similar results. **B** Cross-section of a *C. alba* ovary indicating the trilocular architecture. Orange lines represent a cross-section of the modelling polyhedron. Red arrows indicate the edges of the polyhedron representing inner and outer wall of a carpel. The outer-wall positions overlie the carpel midveins. **C** The ovaries are modelled as polyhedra composed of triangular prisms topped with a truncated pyramid. Red edges denote the inner and outer walls of the same carpel highlighted in (**B**). The thicker lines in (**C**) to (**F**) indicate the base of the style. **D** Two-dimensional projection of the lateral view of a collection of polyhedra. **E** The selection process of the best fitting polyhedron for modelling the ovaries of *C. alba*, consists of choosing among all possible polyhedra based on area, perimeter, and length and width of the ovary. (**F**) Biophysical parameters (spring constants and lengths) of our bead-spring models are optimised for each ovary, such that the equilibrium shape optimally resembles the corresponding best fitting polyhedron. Internal springs, corresponding to the dashed lines in (**C**), are indicated in grey. **G**, **H** Elongation of the inner and outer walls of the *C. alba* ovaries (**G**) and stiffness of the organ walls (**H**) as predicted by our mechanistic model. Stiffness is shown in the vertical (v) and horizontal (h) directions. Dots indicate individual predicted values and bars the mean. $n = 6$.

each ovary the "target" polyhedron whose two-dimensional side view most accurately represented the ovary in its later stages of development (see Fig. 3B–E, and Figs. S3, S4). The resulting solid figures were skewed at the top, reflecting the differential carpel elongation responsible for stylar deflection. Second, we searched for the parameters that most closely reproduced the target polyhedron as a bead-

spring system at its dynamic equilibrium, with the polyhedron edges as springs (Fig. 3F). We allowed different elongation (increase in resting length) and stiffness in the springs on the carpel 1 (Fig. 4A; the adaxial carpel opposite the direction of style deflection) vs carpel 2 (the adaxial carpel on the side to which the style deflects) and carpel 3 (the abaxial carpel) sides compared to a symmetric polyhedron based on

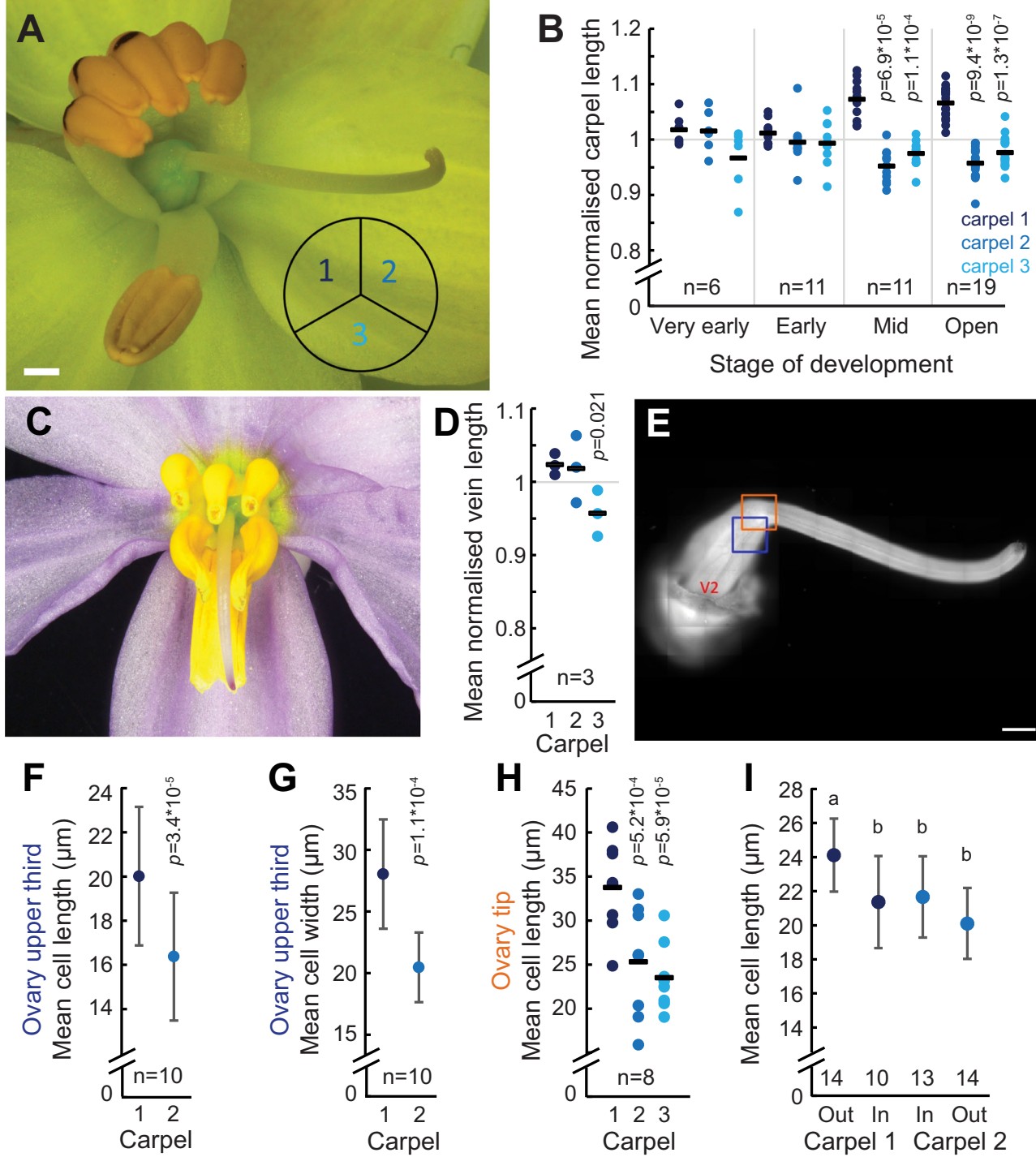

**Fig. 4 | Differential carpel elongation underlies style deflection in *Cyanella alba* subsp. *flavescens*. A** An open *C. alba* flower. The arrangement of the three carpels in the trilocular ovary is indicated by the subdivided circle. **B** Carpel lengths from ovaries of flowers at the indicated stages of development. Lengths were normalised to the average value of the three carpels per ovary. Dots here and in (**D, H**) indicate individual measurements and bars the means. Sample sizes are indicated and show the number of ovaries from which the paired measurements were taken. *p*-values here and in (**D–H**) are from paired two-sided *t*-tests against carpel 1. If no *p*-values are indicated, they were larger than 0.05. **C** *C. hyacinthoides* is a closely related species without enantiostyly. The style is deflected downwards and placed centrally between the six anthers. **D** Measurements of carpel lengths from *C. hyacinthoides* ovaries. **E** Fluorescence micrograph of a calcofluor white-stained pistil. Blue and orange boxes indicate the regions where the measurements of cell sizes shown in

(**F, G**) and (**H**) were taken, respectively. V2 indicates midvein of carpel 2. **F, G** Cell lengths (**F**) and cell widths (**G**) of epidermal cells overlying the midveins of carpels 1 and 2 in the upper third of the ovary are shown. Cell length is the extension of the cell along the long axis of the ovary, and cell width the extension perpendicular to the long axis of the ovary. Values are means ± SD. **H** Cell lengths of epidermal cells overlying the midveins of carpels 1 and 2 at the tip of the ovary are shown. **I** Lengths of subepidermal cells in the outer and inner walls of carpels 1 and 2. The number of carpels from which the measurements were taken is indicated inside the graph. Values are means ± SD. Letters indicate mean cell lengths that are significantly different, as determined by a two-way repeated measures ANOVA followed by Tukey's HSD post-hoc test ($\alpha = 0.05$). In total 14 ovaries were imaged, but individual walls could not be imaged in sufficient quality for some. Scale bars are 1 mm.

very-early bud measurements (Fig. S5A, B, Table S4 and Supporting Text). We used random search optimisation algorithms[38], to identify the parameters of our mechanistic model that best reproduced the target polyhedron and, hence, of the ovary of *C. alba* during the late developmental stage (Figs. 3, S5, S6, Table S6).

The results of this optimization indicated that both the inner and outer walls of carpel 1 undergo more significant elongation than those of carpels 2 and 3 (Fig. 3G). The slight elongation seen in carpels 2 and 3, as predicted by our model, arises from the tensile forces generated by links between different carpels at the base of the style. This effect is driven by the greater relative stiffness of the style base compared to other system walls (Fig. 3H) and contributes to guiding the shape and morphogenesis of the style[39,40]. Indeed, running our model with lower stiffness values generates stylar bending that far exceeds the experimentally observed range of realistic deflection angles (Table S8). Our analysis revealed that both inner and outer walls of carpel 1 end up being compressed (actual length shorter than spring resting length), whereas those in carpels 2 and 3 are stretched (Table S7). Based on the analyses of six *C. alba* ovaries through our models, the primary driver behind the elongation of carpel 1 is the extension of the outer wall, with an average detected elongation relative to the initial length of the early bud stage of 2.47 (SD = 0.41) times, as opposed to the inner wall mean relative elongation of 1.28 (SD = 0.18) times (Fig. 3G).

The above model makes two key predictions. First, the outer wall of carpel 1 should elongate more than the outer wall of carpel 2. Thus, in an R-flower we would predict the top left carpel (when viewed from the front) would elongate more than the top right carpel (Fig. 4A). The second prediction is that the outside wall of carpel 1 should elongate more than the inner wall corresponding to the placenta. To test the first prediction, we measured the lengths of the three carpels in the ovary along their prominently visible midveins at four different stages of development and combined data for L- and R-flowers. Carpel 1 was significantly longer than carpel 2 and the abaxial carpel 3 in both open flowers and mid-stage buds (Fig. 4B). Carpel 2 was on average 89.9% ± 4.3% and 88.9% ± 5.42% of the length of carpel 1 in open flowers and mid-stage buds, respectively. By contrast, carpel lengths were indistinguishable in early or very early-stage buds (Fig. 4B). This mirrors the observation that clear deflection of the style to the left or right is only visible in mid-stage buds and open flowers and supports the hypothesis that differential elongation between the two adaxial carpels causes style deflection in *C. alba*.

To further support the link between differential carpel elongation and style deflection, we repeated this experiment on *C. hyacinthoides*, a close relative of *C. alba* that does not exhibit enantiostyly (Fig. 4C). Styles of *C. hyacinthoides* are not deflected to the left or right, but instead towards the abaxial side. The structure of the trilocular *C. hyacinthoides* ovary matches that of *C. alba*. Carpels 1 and 2 of *C. hyacinthoides* were indistinguishable in length, but carpel 3 was significantly shorter than carpel 1 (Fig. 4D). This suggests that the reduced expansion of carpel 3 underlies the downward deflection of the style in *C. hyacinthoides*, whereas the differential elongation of carpels 1 and 2 determines the LR deflection of the style in *C. alba*.

We next asked whether the greater elongation of carpel 1 in *C. alba* ovaries, as seen above from measuring the length of the carpel midveins, was solely due to more cell elongation or was also accompanied by more cell division. The length of the epidermal cells overlying the carpel midveins was measured from confocal laser-scanning microscopy images of the distal third of ovaries (Fig. 4E, Fig. S7). Cell length (i.e. the size of the cell along the ovary's longitudinal axis) was significantly greater along the midvein of carpel 1 than along carpel 2, with mean cell length in carpel 2 only 81.8% of that observed in carpel 1 (Fig. 4F). This difference is more than sufficient to explain the observation that carpel 2 length is on average 89.9% of carpel 1 length (Fig. 4B). In addition, carpel 1 cells are significantly wider than carpel 2 cells (Fig. 4G). To confirm the observed differences in cell length, we

repeated the measurement at the tip of the ovary where it transitions into the style. Again carpel 1 cells were significantly longer than carpel 2 (and also carpel 3) cells, with the difference between carpels 1 and 2 even greater (25%) than in the upper third of the ovary (Fig. 4H), suggesting a proximo-distal gradient in differential cell elongation.

To test the second prediction from our model, we compared cell lengths between the outer carpel walls and the inner walls at the placenta for carpels 1 and 2 (Fig. S8). Subepidermal cells were used to test this prediction, as they could be clearly and reproducibly measured, and the epidermis was often damaged due to the need to remove the ovules before imaging. As expected, outer-wall cells from carpel 1 were longer than outer-wall cells from carpel 2, confirming the above measurements. Outer-wall cells from carpel 1 were also longer than inner-wall cells from carpel 1, while those of carpel 2 were not (Fig. 4I). Thus, these measurements fully support the predictions from the model.

Our model considers differential carpel elongation in the ovary as the main driver for style deflection. It is conceivable that such differential elongation also occurs within the style itself and contributes to its deflection. Addressing such a contribution would require measuring the length of the style along the three carpels; however, both the measurements and their interpretation would be confounded by the upward bend of the style at its tip (Fig. 1A). Differential elongation of the style part of the three carpels (ignoring the upward bend at the tip) should result not only in deflection of the style away from the midline, but also in its curvature. To assess this possibility, we defined three points (style/ovary transition, beginning of upward bend at the style tip, and mid-point between these) and measured the angle between the two lines connecting these points (Fig. S9). At 175° and 176° these angles were very close to the 180° of a perfectly straight style in both left- and right-handed flowers (Fig. S9). Thus, although we cannot exclude a contribution of differential carpel elongation within the style to overall style deflection, such a contribution - if any - appears to be small relative to the role played by differential growth in the ovary.

Collectively our empirical results and biophysical model indicate that stronger isotropic cell expansion in the outer wall of carpel 1 than carpel 2 drives style deflection to the opposite side of the midline from carpel 1 resulting in L or R stylar deflection.

## Differential carpel growth is associated with increased expression of auxin-induced and elongation-related genes in carpel 1

Given the differential elongation of the two adaxial carpels, we compared their gene expression patterns using RNA-seq data. Outer walls of carpel 1 and 2 were dissected from early- and mid-stage buds with their individual directions of style deflection clearly discernible. We identified 53 differentially expressed transcripts (FDR < 0.05, absolute log2 fold-change(carpel 1/carpel2) > 1). These included four *Auxin/Indole-3-Acetic Acid* (*Aux/IAA*) and one *SMALL AUXIN UP-REGULATED* (*SAUR*) transcripts, all of which were upregulated in carpel 1 (Supplementary Data 2). To complement this individual-transcript based analysis, we functionally annotated transcripts using Mercator4[41] and tested for bins that differed significantly from the remaining transcriptome in terms of their distribution of log2 fold-change values. In total, 106 bins showed a significantly different distribution compared to the background (Supplementary Data 3). The empirical cumulative-distribution function (ECDF) plots indicated that for many of these bins gene-expression differences between the two carpels were less variable than the transcriptomic background, consistent with these being large bins of house-keeping genes (Fig. S10). We identified three bins with a possible link to cell elongation as significantly different from the background. These were "Cell wall organisation", "Phytohormone action" and "Cytoskeleton organisation". The most specific significant subcategories within these were "pectin methylesterase" and "Fasciclin-type arabinogalactan protein"; "RALF/RALFL-precursor polypeptide", "auxin-responsive genes *(SAUR)" and "transcriptional

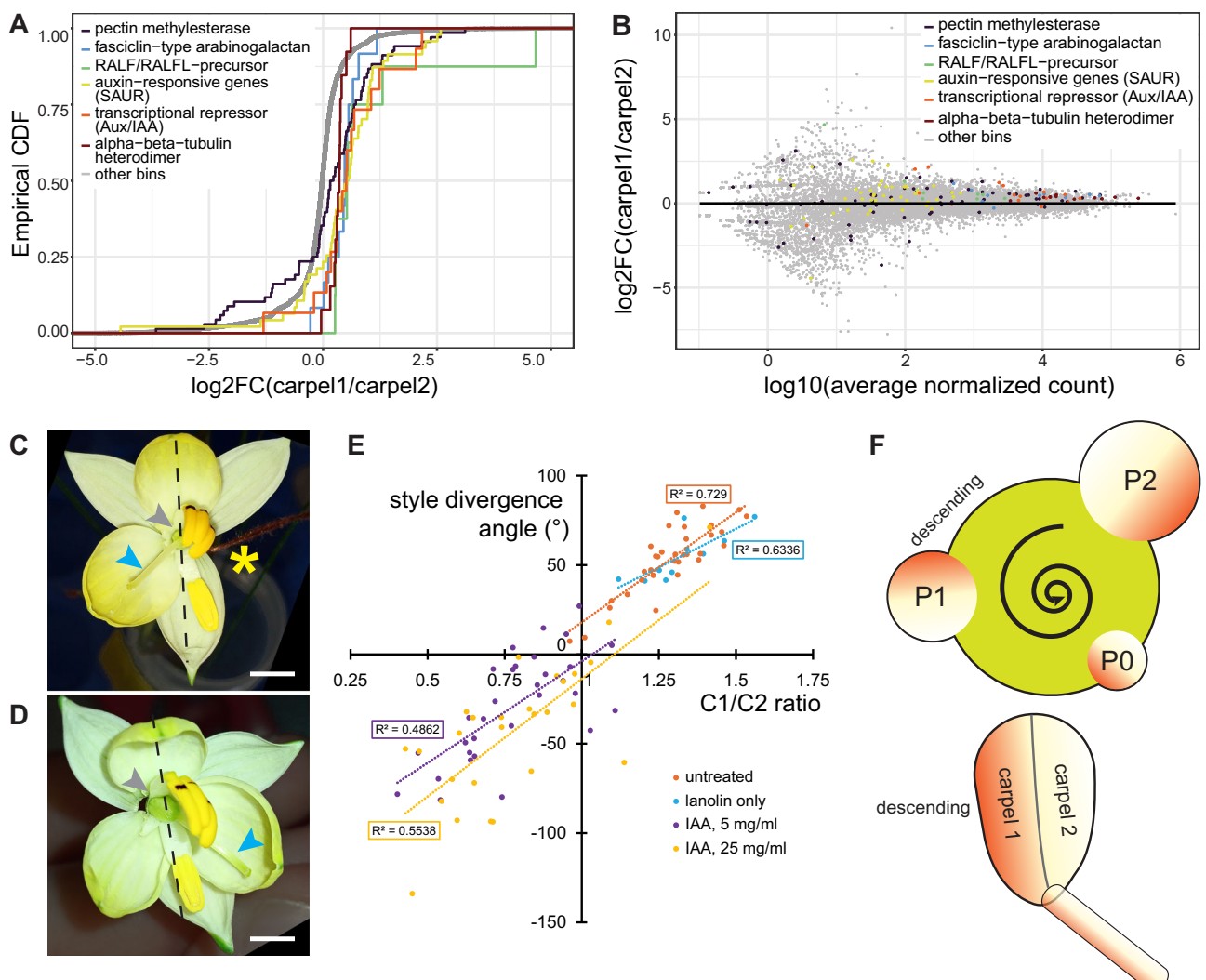

**Fig. 5 | Differential auxin signalling between the two adaxial carpels underlies style deflection in *Cyanella alba* subsp. *flavescens*.** **A** Empirical cumulative distribution functions (ECDFs) of the log2 fold-changes (log2FC) in transcript abundance between carpels 1 and 2 are shown for selected Mercator bins with significantly different log2FC distributions compared to the transcriptome-wide background ('other bins'). See Supplementary Data 3 for results for all bins. **B** Log2 fold-changes (log2FC) in transcript abundance between carpels 1 and 2 are shown relative to the average base expression across all samples for all genes in the Mercator bins shown in (**A**). **C**, **D** *C. alba* flowers from left-biased plants after treatment with lanolin only (**C**) or lanolin with 5 mg/ml IAA (**D**) to carpel 2. Grey arrowheads indicate the side where two of the adaxial anthers were removed and lanolin was applied to the carpel 2. Blue arrowheads indicate styles. Dashed lines show the midline of the flower based on the adaxial petal and stamens. One of the three petals in (**C**) was removed (yellow asterisk). Scale bar is 5 mm. **E** Effect of auxin on style divergence angle and the ratio of carpel 1 to carpel 2 length (C1/C2 ratio). Positive divergence angles indicate style deflection away from carpel 1, negative angles towards carpel 1. Linear regression lines and R2 values are shown. Both IAA treated samples are significantly different from untreated and lanolin-only samples regarding both C1/C2 ratio and style divergence angle ($p < 0.001$; $t$-test). **F** Schematic model for the development of a consistent floral handedness based on an initial asymmetry in auxin concentration in the initiating organs (gradients in circles, P2 to P0 from old to young; descending indicates the side of the primordium facing the next older primordium) at the shoot meristem (green). The direction of this initial asymmetry depends on the direction of the phyllotactic spiral (curved arrow, counter-clockwise in this illustration) and is later translated into style deflection (to the right in this illustration), by stronger elongation of carpel 1 than carpel 2.

repressor *(Aux/IAA)"; and "alpha-beta-tubulin heterodimer", respectively. For each of these, transcript levels were higher in the carpel 1 than carpel 2 samples, visible by a consistent shift in the ECDF plot to positive values (Fig. 5A). This higher expression in carpel 1 was particularly evident when considering those members of each gene family with higher absolute expression levels (Fig. 5B). As discussed below, the elevated expression of transcripts encoding pectin methylesterases, Fasciclin-type arabinogalactan proteins, SAUR and Aux/IAA proteins in carpel 1 is consistent with increased cell elongation driven by higher auxin levels in carpel 1 versus carpel 2. Our transcriptome analysis therefore suggests a molecular basis for the differential growth of the two adaxial carpels driving style deflection.

## Auxin application to carpel 2 reverses the style orientation

To test the importance of differential auxin signalling between carpels 1 and 2 we asked whether external auxin application to carpel 2 could lead to straight styles or even a reversal of style orientation by causing carpel 2 to expand as much as, or more than, carpel 1. Lanolin with or without auxin was applied to the outside wall of carpel 2 in early-stage flowers (Fig. 5C, D). After the flowers had opened, we dissected out the pistils and measured the lengths of carpels 1 and 2 along their midveins as well as the degree of style deflection relative to the long axis of the ovary. Across all treatment groups, there was a positive correlation between the ratio of carpel 1/carpel 2 lengths and the degree of style deflection; thus, the more uneven the expansion of the two carpels, the

greater the deflection of the style is (Fig. 5E; positive versus negative angles indicate styles deflecting away from carpel 1 versus towards carpel 1, respectively). However, while all untreated or lanolin-treated pistils had carpel 1/carpel 2 ratios above 1 and styles deflected away from carpel 1 (positive angles), application of auxin to carpel 2 shifted both measures towards lower values ($p < 0.001$ for all comparisons of auxin treated to control groups based on $t$-test). This indicates that auxin application could equalize growth of the two carpels or even cause carpel 2 to be longer than carpel 1. As predicted, this led to the style being straight or even deflected away from carpel 2 and towards carpel 1. Together with the transcriptomic analysis, this indicates that in untreated plants higher auxin signalling in carpel 1 causes it to expand more than carpel 2, leading to style deflection away from carpel 1. Thus, unequal auxin-induced expansion of the carpels appears to be the primary cause of floral handedness in *C. alba*.

## Discussion

Our results provide multiple lines of evidence indicating that *C. alba* possesses monomorphic rather than dimorphic enantiostyly. Molecular analyses revealed that the consistent handedness of the mirror-image flowers of individual plants within any single flowering season is not determined by their genotype. Rather, style deflection depends on the apparently random choice that each annually initiated shoot meristem makes concerning the handedness of the phyllotactic spiral. Once determined, the handedness of the phyllotactic spiral is stable throughout one growing season, which translates, albeit with occasional deviations, to a predictable and consistent direction of style and anther deflection per individual. We also show that this left-right asymmetric morphology has a simple cellular and developmental basis. Developmentally, style deflection is caused by differential cell elongation between the two adaxial carpels, coupled with a very stiff base to the ovary as inferred from our biophysical model. The stronger elongation of one of the carpels results from increased auxin signalling and higher expression of cell-expansion promoting genes. Thus, our work identifies a non-genetic mechanism for ensuring consistent left-right handedness that links chirality across different levels of plant development to produce adaptive morphological variation.

The mechanism for determining stylar deflection based on the handedness of the phyllotactic spiral mirrors recent findings about leaf asymmetries in other plants. In tomato, *Arabidopsis thaliana* and several other species, the direction of more or less subtle LR asymmetries in leaves can be predicted from the handedness of the phyllotactic spiral[11,16,42]. These asymmetries have been traced back to an unequal distribution of auxin and auxin signalling in initiating leaf primordia at the shoot apical meristem[16]. In particular, the side of the primordium facing the next oldest (the so-called descending side) has higher auxin levels than the ascending side, resulting from the pattern of polar auxin transport. We therefore consider it highly likely that floral asymmetry in *C. alba* is ultimately caused by such an initial auxin asymmetry in initiating bract/floral primordia (Fig. 5C) for two reasons. First, auxin maxima seem to determine the sites of organ initiation at the shoot apical meristem across all mono- and dicotyledonous plants studied[43], so most likely also in *C. alba*. Second, the asymmetry of auxin levels in initiating primordia in spiral phyllotaxis appears to result from the patterning mechanism of polar auxin transport itself that generates these auxin maxima, leading to what has been interpreted as an inherent mechanistic constraint on the starting conditions in an initiating primordium[16], which most likely applies to all species with spiral phyllotaxis.

This model raises three questions. How is the presumed initial auxin asymmetry translated into floral handedness? Is a similar mechanism operating in other plants with mirror-image flowers? What could be the functional significance of linking floral variation to the working of a core shoot-patterning mechanism? In a *C. alba* individual with counter-clockwise phyllotaxis and style deflection to the right, the left adaxial carpel (when viewed facing the flower) corresponds to the descending side of the initial primordium. Thus, according to our model the stronger auxin signalling on the descending side of the primordium ultimately triggers stronger cell expansion in the corresponding adaxial carpel (Fig. 5C). When combined with a very stiff base of the ovary, such differential carpel elongation is predicted by our biophysical model to cause style deflection away from the descending side of the flower. Strikingly, even though we cannot directly measure auxin distribution in *C. alba* due to the lack of a suitable reporter, our transcriptomic comparison between the more- and less-expanding adaxial carpels provides evidence for stronger auxin signalling in the carpel on the descending side of the initial primordium. This carpel has higher expression of Aux/IAA and SAUR encoding genes, both of which are induced by auxin signalling[44,45]. The latter also have a well-established role in mediating acidification of the extracellular space, resulting in increased cell expansion as described by the acid-growth hypothesis[44,46,47].

In addition, the more strongly expanding adaxial carpel also shows higher expression of genes encoding extracellular pectin methylesterases and Fasciclin-like arabinogalactan proteins (FLAs). Overexpression of a pectin methylesterase in *A. thaliana* shoot meristems resulted in less rigid cell walls[48], presumably facilitating cell expansion, while overexpression of a FLA in cotton-fibre cells increased their elongation[49], suggesting that both activities also contribute to stronger expansion of the adaxial carpel on the descending side of the flower. Auxin and the auxin-response factor ARF3/ETTIN have been shown to induce pectin methylesterase activity or expression in tobacco and *A. thaliana*, respectively, suggesting a similar link in the descending carpel[50–52]. At the same time, this carpel also shows higher expression of potential negative regulators of cell expansion, in particular Rapid Alkalinisation Factor (RALF) peptides and three pectin-methylesterase inhibitors, which limit cell expansion in *A. thaliana* roots and shoot meristems, respectively[48,53]. RALF signalling has been shown to increase auxin biosynthesis in *A. thaliana* roots[53], providing another possible link between the differentially expressed gene categories in the *C. alba* carpels. The importance of differential auxin signalling between the two carpels is further supported by our external auxin application to carpel 2, which was sufficient to equalize growth between the two carpels or even lead to carpel 2 overgrowth, causing the style to become straight or deflected towards carpel 1.

Thus, even though the mechanistic details remain to be resolved and auxin levels have not been directly quantified, our findings suggest that increased auxin signalling in the more expanding adaxial carpel causes stronger cell elongation than in the less expanding one, leading to style deflection away from the more expanded side of the ovary. At the same time, it remains an open question how the positional information from the presumed initial auxin asymmetry in the primordium could be maintained for the long time across many cell divisions between floral-meristem initiation and late stages of pistil morphogenesis. An epigenetic process may be involved, but this remains speculative at present. However, this long duration between the presumed initial trigger and the later elaboration of the morphological asymmetry may explain the small fraction of 'developmental deviations' where individual flowers do not follow the pattern of style deflection predicted from the handedness of the phyllotactic spiral.

We propose that our model for *C. alba*, which resembles that proposed for leaves[16], may be more widely applicable to other species with floral asymmetry. An interaction between processes at the inflorescence meristem and flower development predicting the handedness of mirror-image flowers is also seen in other species with monomorphic enantiostyly, such as *Solanum rostratum*[54]. The inflorescence of this species is a scorpioid monochasial cyme, i.e. each meristem terminates in a flower and inflorescence growth is continued by a single lateral meristem, whose position alternates relative to the respective main axis. Styles are almost always deflected towards the

lateral branch, resulting in pendulum asymmetry[55], and suggesting that positional information from the primordium stage translates into floral handedness. A more directly comparable case is found in *Monochoria australasica* (also referred to as *Pontederia australasica*). This species of Australian perennial marsh plant shows inflorescence-level monomorphic enantiostyly, i.e. the flowers within one inflorescence have a consistent handedness, yet different inflorescences produced by the same individual can have a different floral handedness[24]. Inspection of publicly available photographs of *M. australasica* inflorescences shows that the species has spiral phyllotaxis (Fig. S11, Table S9). Across all available images where the handedness of both phyllotaxis and style deflection could be reliably assessed, we found that 7 of 9 inflorescences with a clockwise phyllotactic spiral had styles deflected to the right (one inflorescence mixed, one with L-flowers), and all 12 inflorescences with counter-clockwise phyllotaxis had styles deflected to the left. Thus, the handedness of the phyllotactic spiral again appears to determine floral handedness, although with an inverted relation compared to *C. alba*, suggesting that the mechanism identified here also operates in other systems with monomorphic enantiostyly.

In plants with monomorphic enantiostyly where several flowers with different handedness are open on a given day, there is a risk of between-flower selfing (geitonogamy), as approximately half of the flowers should donate pollen efficiently to the other half[21,27]. This could result in inbreeding depression and gamete wastage due to pollen discounting. Linking the handedness of individual flowers to phyllotaxis as a stable pattern-generating process at the shoot apical meristem can unify floral handedness across an inflorescence, thus limiting the mating costs of geitonogamy[56]. At the same time, the random establishment of clockwise versus counter-clockwise phyllotaxis across individual shoot meristems results in an equal distribution of L- and R-biased inflorescences in a population, maximizing mating opportunities and effectively functioning like dimorphic enantiostyly. These beneficial effects would be maximal in a situation where the relation between phyllotactic and floral handedness was absolute, individuals only formed a single inflorescence, and there was a large floral display with many open flowers on a given day. The latter seems to be the case in *M. australasica*, yet its clonal growth with the possibility of forming more than one inflorescence per individual reintroduces some risk of within-plant selfing owing to between-inflorescence geitonogamy. In *C. alba*, each plant appears to form only a single inflorescence per growing season and daily floral display size is small, but often larger than one. This suggests that inflorescence-level uniformity of floral handedness can limit within-plant selfing and promote disassortative mating between individuals of opposite phyllotactic and floral handedness.

Although enantiostyly is a relatively rare phenomenon, our study highlights how a prominent phenotypic polymorphism in nature that has been known since Darwin's studies on plant sexuality[25] can be based on a random event during development. The random 'choice' of the direction of phyllotaxis when a new meristem is initiated is combined with an inherent developmental constraint in a core developmental process, the polar auxin transport leading to the auxin maxima that trigger the initiation of new primordia. This constraint leads to chirality, with the left versus the right sides of the primordia differing consistently from each other. Ultimately these differences are translated into ecologically relevant morphological variation in the form of LR asymmetry in flowers. Our study underscores how the investigation of uncommon systems can offer additional perspectives on general principles of development and their impact on ecology and evolution.

## Methods

### Plant study system and sites
*Cyanella alba* subsp. *flavescens* is a narrow endemic native to the Western Cape of South Africa. It is restricted to the northern Cederberg and Olifants River region and is locally abundant in the Biedouw River Valley and at Wupperthal (see Figure 15 in Manning and Goldblatt, 2012). The species is a long-lived, deciduous geophyte with a deep-seated corm and pale-yellow enantiostylous zygomorphic flowers with styles deflected to the left and right side of the flower (see Fig. 1A in main text). Anthers are dimorphic (heteranthery) with poricidal dehiscence. The upper five centrally placed "feeding anthers" are fused and the sixth "pollinating anther" is deflected downwards to either the left or right side of the flower in an opposite direction to the style.

Populations in the Biedouw River Valley and Wupperthal are large (estimated at 2000-5000 individuals) and flower from early August to October after winter rains. The three populations we sampled for demographic studies (see below) in the 2022 and 2023 flowering seasons were located in the Biedouw Valley (site 1 at 32°08'18" E19°10'57", 400 m a.s.l and site 2 at S32°11'25" E19°10'17", 560 m a.s.l.) and Wupperthal (site 3 at S32°16'17" E19°12'42", 530 m a.s.l.). All sites occurred on open degraded pasture comprised of renosterveld vegetation and all sites were separated by a minimum of -10 km. Our research was approved by Cape Nature (Permit CN35-87-25844) and we collected plants with permission from Mr Barry Lubbe of Mertenhof farm and Barend Salomo of the Wupperthal Original Rooibos Cooperative.

### Demographic study
To investigate the dynamics of floral handedness in *C. alba* subsp. *flavescens* throughout two flowering seasons, we marked 75 plants at each of the three sites with a unique number at the beginning of 2022. The number was etched into a metal tag to withstand weathering and secured next to the plant using a 10 cm nail driven into the ground. We recorded the handedness of flowers on each plant across all three sites every one to two days during the 2022 flowering season, and every four to five days during 2023. At each visit, newly opened flowers were recorded as L or R and marked individually with a jeweller's tag. Flowers of *C. alba* subsp. *flavescens* remain open on average for 8.9 days, range 5–12 days[31]. In 2023 we modified our methods for plants with more than one flower open by recording the position of the flowers within the inflorescence and assigning those closest to the ground as having opened first. This more accurately estimated the sequence of handedness in the inflorescence. We also recorded for all plants the number of buds, open and wilted flowers at each visit.

To identify the hypothesized *E*-locus controlling L- or R- handedness in 2022 we collected two to three leaves from 16 individuals from sites 1 ($n = 8$), 2 ($n = 5$), and 3 ($n = 3$) that produced six flowers that were either R or L only. The leaves were dried in silica before gDNA extraction and sequencing as described below.

### DNA extraction and sequencing
We extracted total genomic DNA from silica dried or frozen leaf material using Nucleobond High Molecular Weight DNA extraction kits (Macherey-Nagel™, cat no.: 740_160.20) and samples were sent to Novogene UK (Cambridge Sequencing Centre) for whole genome sequencing on the Illumina platform (NovaSeq 6000). A single, right-handed sample (C_209) was submitted to Novogene UK for high fidelity PacBio sequencing (PacBio sequel II/IIe DNA HiFi library).

### RNA extraction and sequencing
We extracted RNA from *C. alba* subsp. *flavescens* pistils, including the style and half of the ovary, at three developmental stages: "mid", "early" and "very early" using a basic phenol:chloroform extraction method followed by purification with an RNA Miniprep kit (Zymo Research: Direct-zol RNA Miniprep Kits; cat. no. R2050). We assigned buds to bud classes based on a number of visual criteria detailed in Fig. 3A, and all open flowers and dissected buds on an individual plant were examined before assigning a handedness phenotype (L or R) to a

sample. We pooled dissected pistils at each developmental stage across multiple individuals to generate three biological replicates. Mid and early-stage samples consisted of five to seven pistils (~30 mg tissue) and very early samples consisted of ten pistils (~20 mg tissue). RNA was sequenced on the Illumina platform following Eukaryotic mRNA library construction (polyA enrichment).

We also extracted RNA from the outer walls of carpel 1 and carpel 2 of *C. alba*. We divided the individuals into four groups and pooled the dissected outer walls of carpel 1 and carpel 2 in each group separately to generate four pairs of biological replicates. Each sample consisted of ~15 carpels. The RNA was extracted and sequenced using the same methods described in the preceding paragraph.

## PacBio assembly

We used the *k*-mer counting tool KMC version 3.2.1[57] to count *21*-mers in the PacBio sequences of the sample C_209. Then we used GenomeScope2.0[58] to estimate the genome size and ploidy of *C. alba* subsp. *flavescens*.

The assembly was generated using hifiasm version 0.19.0-r534[59] and we used default settings with only "--hg-size" (estimated haploid genome size) specified. Haploid genome size estimates of 1 G and 700 M were used, in line with flow cytometry and GenomeScope2.0 size estimates, and we estimated completeness and contiguity of the assemblies using BUSCO version 5.4.4 and QUAST version 5.2.0[60,61]. The database chosen for the BUSCO analysis was liliopsida_odb10. The "-m genome" and "--augustus" options were specified.

We downloaded the *Arabidopsis* protein database "Araport11_pep_20220914" from the website 'The *Arabidopsis* Information Resource' (TAIR, www.arabidopsis.org). We used tblastn to align the *Arabidopsis* protein sequences to the reference genome, to identify protein-coding regions in the genome. Bedtools version 2.30.0[62] was used to merge identified regions that are <10 bp apart.

## Genome Wide Association Study (GWAS)

We used NextGenMap version 0.5.5 to align Illumina reads to the primary PacBio assembly[63]. We used SAMtools version 1.3.1[64] to exclude reads with mapping quality lower than 10, then sort and index the alignment files. We used the function "mpileup" of BCFtools version 1.16 (Danecek et al. [64]) to generate binary variant call files (BCFs) with the alignment files. The "-Q 10" option was specified to skip bases with base quality lower than 10, and only protein-coding regions in the reference genome were included. We used the function "call" of BCFtools to identify potential variant sites with the multiallelic calling model option (-m) included. We used VCFtools version 0.1.16[65] to label genotypes with coverage <3 as "missing", and filter sites with minor allele count lower than 3, mapping quality <30, or >50% "missing" genotypes (options: --minDP 3; --mac 3; --minQ 30; --max-missing 0.5). We used BCFtools to keep only single-nucleotide polymorphisms (SNPs) (option: --types snps). Finally, we used PLINK version 1.9.0 (http://pngu.mgh.harvard.edu/purcell/plink/) to identify SNPs significantly associated with L or R morphs (Cochran-Armitage test, $P < 0.0005$).

## Coverage analysis

We performed coverage analyses with non-overlapping genomic windows of 10 kb. Illumina reads were aligned to the two haplotypes of the genome assembly using NextGenMap and analysed separately. SAMtools was used to remove reads with a mapping quality <10 before sorting and indexing the alignment files. We calculated read coverage over 10-kb windows using mosdepth version 0.3.3[66] and coverage of each window was normalised by dividing by the genome-wide average read coverage. We analysed normalised coverage scores in R version 4.3.0 and *t*-tests were performed to determine whether the normalised coverage scores of left- and right-handed samples were significantly different for each window, with a cut-off of $P < 0.0005$. The ratio of

normalised coverage for L versus R samples was calculated, and filtered for windows where this ratio was > 1.5 or <0.67, indicating lower coverage in one morph compared to the other. We used IGV version 2.14.1[67] to investigate regions of interest.

## KmerGO2

We used KmerGO2 (Wang et al. [68], https://github.com/ChnMasterOG/KmerGO2) to compare the *k*-mer composition of L- and R-individuals to identify *k*-mers characteristic of each morph. We used $k = 40$ and ran subcommands "kmc3" and "filtering" with 24 threads and "union" with 8 threads.

We imported the output of KmerGO2 into Microsoft Excel for further analysis and calculated the total number of morph-associated *k*-mers (Average of Sensitivity and Specificity (ASS) score >0.8) for each dataset. We combined presence/absence scores for each *k*-mer for the forward and reverse fasta file of each individual and the total number of L-associated and R-associated *k*-mers was plotted in Excel.

## Transcriptomic analysis

We assembled a draft transcriptome using the combined reads from the 18 RNA samples of pistils. The de novo assembly was created using Trinity version 2.14.0[69]. The "--max_memory" argument was set to 50 G and the "normalise_by_read_set" parameter included to limit RAM usage. We conducted functional annotation for the draft transcriptome using Mercator4 v6.0[70] with Prot-scriber and Swissprot annotations included. We used the read aligner Salmon (version 1.9.0[71]) with the "--no_bowtie" option and we performed read pseudoalignment and quantification with Salmon. We used the Salmon "quant" command with the "--gcBias" and "--validateMappings" settings. Count data was analysed in R-studio (R version 4.3.0) using the package DESeq2[72]. Graphs were plotted using the R package ggplot2[73]. For each gene, the functional annotation of the longest isoform was selected as the annotation of the gene.

We aligned the RNA-Seq data from 18 pistil samples and eight carpel samples to the reference genome with STAR[74]. Then we conducted the structural annotation of the reference genome using BRAKER3[75], with the aligned RNA reads and the partition Viridiplantae of the protein database OrthoDB v11[76] as evidence. We conducted functional annotation for the coding sequences predicted by BRAKER3 using Mercator4 v6.0[70] with Prot-scriber and Swissprot annotations included. Since *SMALL AUXIN UP-REGULATED* (SAUR) genes were not included in the Mercator bins, we manually created a Mercator bin: 11.2.2.6, "Phytohormone action.auxin.perception and signal transduction.auxin responsive genes *(SAUR)", for genes that were annotated as SAUR genes.

We quantified the expression level of each predicted gene in the eight carpel samples with StringTie[77]. We conducted gene differential expression analysis with the count data using the R package DESeq2. Then we conducted the MapMan enrichment analysis using MapMan4[70], with the results of the gene differential expression analysis and the Mercator4 annotation as the input. For each gene, the functional annotation of the isoform with highest read count across all the samples was selected as the annotation of the gene. Graphs were plotted using the R package ggplot2.

## Measurements of phyllotaxis and data analysis

Plants for the phyllotaxis experiment were sampled at all three sites in the Biedouw valley described above, between 30 August and 9 September 2023.

Angles between subsequent leaf/floral stalk primordia were measured with a hinged protractor, a circular histogram that can be clipped non-destructively around the plant's stem. The circle was divided into 16 equal bins of 22.5°, with bin 1 centered around 0 degrees, and bin numbers increasing in a clockwise (CW) fashion. The angle of the leaf at the base of a floral stalk, when bent outwards, was

used as proxy for primordium orientation. With the older organ aligned to the 0 degree line within bin 1, bin 11 corresponds to the 137.5° angle of a counterclockwise (CCW) phyllotactic spiral, when looking from older to younger organs, as is common in the field. Conversely, bin 7 corresponds to a clockwise golden angle. See Figure S2A. (Note: data was recorded young to old, but reported old to young, the standard of the field. The data was transformed explicitly (swapping bin 7–11, 8–10, etc; Figure S2), or interpreted in opposite direction (bin 11 = CW; Figure 2, scripts + data in[78])).

Histograms of subsequent flower angles were made from all data recorded before September 9. For this analysis, we only filtered against confounded developmental sequences (e.g., containing a bud in an older position than an open flower). All recorded angles are included in the analysis shown in Fig. S2B (88 plants, 247 angles).

As in all but five sampled plants, the phyllotactic spiral was very consistent, i.e., either fully CW or fully CCW, we used a simplified scoring for the *C. alba* plants harvested on September 9: whole plants were scored as CW or CCW, without explicitly measuring the relevant angles. Individual flower orientations were scored as L (left), LU (left-unsure), U (unknown/impossible to determine), R (right) and RU (right-unsure). Where possible, buds were opened to determine flower orientation. If a plant displayed at least one L-flower and no R, the plant's flower orientation was scored as left (-1). In the opposite scenario, it was scored as right (1). In all other cases, i.e., containing both orientations, or no L/R information, the plant was scored as mixed (0). Flowers with unsure style orientation (mostly because they were already senescent or buds were still too young) were excluded from the analysis.

For phyllotactic orientation, the number of CCW and CW angles on a plant were counted. Bin 9 (angles of approximately 180°) was ignored. If both CW and CCW occurred on a single plant, its phyllotactic orientation was coded as unclear (0). We interpret these cases as analogous to the deviations from the regular phyllotactic spiral seen in *A. thaliana ahp6* mutants (so-called M-shapes) where the relative timing of organ initiation is changed, yet the spatial pattern maintained, resulting in three consecutive angles that are much >137.5°[79]. Three of the five plants scored as having unclear phyllotaxis showed a deviation consistent with the beginning of an M-shape in their last measured divergence angle. As many plants had very short internodes (few mm or less), it was sometimes difficult to observe the correct developmental order of organs along the stem, so some of these observations could also have been caused by measurement errors. For similar reasons, plants on which the recorded sequence of generative organs contradicted normal developmental order (e.g., containing a bud in an older position than an open flower), were excluded from the filtered data set. Otherwise, all CW plants were coded 1 and all CCW plants were coded -1. In case relative angles of <90 degrees were recorded (bins 1–4 and 14–16), the phyllotactic pattern was considered confounded and these plants were not considered in the filtered data set. Filtering / preprocessing was done using a custom-built python script available in[78].

Analysis of deviations from prediction relative to flower position was done using a custom built python and R scripts available in[78]. Correlation coefficients and 95% confidence intervals were calculated using the function corr.test in R.

Based on the observed strong correlation between CW phyllotaxis and left-handed flowers, we predicted the flower orientation of all plants with consistent (fully CW or fully CCW) phyllotaxis. Per flower position, with 1 as the oldest primordium, the fraction of deviations (R on CW or L on CCW) was recorded. Confidence intervals for the fraction of deviations per position were calculated based on a binomial test. The baseline of uniform probability of deviations was computed as NumberOfFlowersThatDeviate / TotalNumberOfFlowers Considered. Overall departure from this uniform prediction was tested using $X^2$ (chisq.test in R).

## Stereomicroscopy

We imaged dissected pistils from three different angles to obtain images of the three carpels that make up the ovary using a Nikon SMZ1500 Stereomicroscope fitted with a Nikon Digital Sight DS-Fi2 camera.

## Fluorescent microscopy

**Fixation.** We fixed whole pistils in 4% (w/v) paraformaldehyde (PFA) prepared in 1X phosphate buffered saline (PBS: 136.89 mM NaCl, 2.68 mM KCl, 5.37 mM $Na_2HPO_4$, 1.76 mM $KH_2PO_4$; pH7.4) for 1 h at room temperature. Fixative was removed by rinsing three times in 1X PBS for 5 minutes, and once in $dH_2O$.

**Clearing.** We removed endogenous pigments using the ClearSee protocol for pistils described in Kurihara et al. (2015), with samples incubated in ClearSee (10% (w/v) Xylitol; 5% (w/v) sodium deoxycholate and 25% (w/v) Urea in $dH_2O$) for 6 weeks, or until transparent. The solution was changed three times per week until tissues had cleared.

**Cell wall staining.** Pistils were rinsed in $dH_2O$ for 1 h before being transferred into Calcofluor white solution (Sigma, product no. 18909) for overnight staining. Samples were de-stained overnight in $dH_2O$. All staining and de-staining steps were performed in the dark with gentle agitation. Samples were placed on long coverslips (25 × 50 mm) in water for imaging.

**Imaging.** Cell lengths of epidermal cells overlying the midveins of carpels 1 and 2

We performed fluorescent microscopy using a Zeiss LSM 880 Confocal and overview images of the entire pistil were taken using tile scanning on a widefield Zeiss monochrome CCD camera. We selected two positions on the ovary to take detailed z-stacks of cell structure. The first position was within the top third of the ovary, centred around one of the midveins that runs along the surface of each carpel. The second position was at the ovary-style transition, at the base of the style. The approximate positions of each z-stack were marked by eye on the overview image and the sample was then flipped using tweezers to image the opposite side, and the process repeated.

**Imaging.** Cell lengths of subepidermal cells in inner and outer walls of carpels 1 and 2

To obtain images of the inner walls of *C. alba* ovaries, we removed the adaxial surface of the ovary. Working under a dissecting microscope, we made longitudinal cuts along the prominent midveins of carpels one and two, as well as along either side of the indentation between the two adaxial carpels (i.e. along the carpel wall that separates carpel 1 from carpel 2). A final transverse cut along the base of the ovary was made and the two triangular pieces of tissue removed to reveal the chambers of the adaxial carpels. The ovules were removed by scraping along the placental wall with the tip of a scalpel blade.

Images of the dissected pistil were captured on a Zeiss LSM 880 Confocal microscope under 10X magnification, before taking higher resolution 20X Z-stack images of the cell walls of carpel 1 and carpel 2 at the same relative position.

## Image analysis and processing

All image analysis was performed using the freely available online software ImageJ (https://ij.imjoy.io/).

**Carpel lengths.** We measured carpel lengths using the spline tool. Each carpel was measured three times and the average recorded. Scale bars in each image were also measured in triplicate, using the straight-line measurement tool, and used to convert the length of the carpel in pixels to a measurement in micrometres. To account for variations in

 

flower size, we normalized carpel lengths of individual samples against ovary size using the equation:

$$normalised\ length\ of\ carpel\ x = \frac{length\ of\ carpel\ x\ (\mu m)}{\left(\sum_1^3 length\ of\ carpel\ i\right)/3} \quad (1)$$

**Cell lengths**

**Upper third of the ovary.** We chose an image from each z-stack that clearly showed the outlines of a line of cells running directly above each midvein. We then measured cell lengths of ten cells using the straight-line tool in ImageJ. We measured scale bars in each picture in triplicate and these were used to convert the mean cell length measurements in pixels to measurements in micrometres.

**Base of the style.** Z-stacks captured the ovary-style transition. For each carpel of the style, we chose a line of cells approximately 100 μm from the central crease of the style. An image which clearly captured a line of five or more cells in this position was exported and cell lengths were measured in ImageJ, as described above. We calculated averages over five cells rather than ten.

**Cell widths.** Using the same images as for *Cell lengths: Upper third of the ovary*, we measured cell widths of five cells at 90° to the previous measurement. Cell measurements were performed as previously described.

**Carpel outer and inner cell walls.** An image from the z-stack with the clearest resolution was selected to quantify the lengths of 10 adjacent cells in the subepidermal layer of the outer and inner walls of carpels 1 and 2. These measurements were repeated three times for each image.

**Auxin treatment**

Indole acetic acid (IAA; Sigma-Aldrich) was dissolved in absolute ethanol to produce a stock solution containing 100 mg/mL IAA. Lanolin containing auxin was prepared by melting 0.5 g of lanolin at 40°C and adding IAA stock solution to give a final concentration of either 5 mg/mL or 25 mg/mL IAA. After the IAA addition, the paste was mixed with a pipette tip, vortexed vigorously, and left to solidify. For the mock treatment, a corresponding volume of ethanol was mixed with lanolin.

*Cyanella alba* subsp. *flavescens* plants were collected from site 1 in the Biedouw Valley and maintained in water with cut flower food (Chrysal: Clear universal flower food). Individuals were selected to have at least one open flower and at least one early-stage bud for treatment. The handedness of the buds was extrapolated from the style deflection of the open flowers. We opened early-stage buds, removed two of the five adaxial stamens overlying carpel 2 and applied a small bead of lanolin paste to carpel 2 using a wooden toothpick. After closing the buds again the plants were kept at room temperature. When the treated flowers opened, all floral organs except the pistil were removed, the midvein of carpel 1 was marked and the pistils were photographed under a dissecting microscope (Olympus SZ61) fitted with a Zeiss Axiocam 208 colour camera from the adaxial side, facing the line where carpels 1 and 2 are fused (see Stereomicroscopy section above). From these images we measured the lengths of midveins 1 and 2 as above. Style deflection was quantified by measuring the angle between a line drawn perpendicular to the base of the ovary and a line drawn through the middle of the style at its base.

**Statistics and reproducibility**

Statistical analysis was performed using R (version 4.3.0) or Microsoft Excel 2016. No statistical method was used to predetermine sample size. Sample sizes were chosen to be as large as logistically feasible. No data were excluded from the analyses. Samples were randomly allocated to experimental groups for the auxin treatment. The

Investigators were not blinded to allocation during experiments and outcome assessment. The correlation between the direction of phyllotaxis and the direction of style deflection was confirmed by sampling in the 2023 and 2024 flowering season. The auxin effect on style deflection was shown to be reproducible based on two independent experiments. In the transcriptomic analysis of styles, each type of tissue had three biological replicates. In the transcriptomic analysis of carpel walls, each type of tissue had four biological replicates. For all other experiments, replicate numbers are given directly in the figures.

**Reporting summary**

Further information on research design is available in the Nature Portfolio Reporting Summary linked to this article.

## Data availability

The whole-genome sequencing and transcriptome datasets generated in this work have been deposited in NCBI under accession code Bio-Project PRJNA1123744. The data for analysing phyllotaxis are available on Zenodo (https://doi.org/10.5281/zenodo.14989473)[78]. All other data generated in this study are provided in the Source Data file. Source data are provided with this paper.

## Code availability

The code for the custom scripts used to analyze phyllotaxis is available on Zenodo (https://doi.org/10.5281/zenodo.14989473)[78].

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

## Acknowledgements

Plant material was collected under Cape Nature permit (CN35-87-25844). We are grateful to Marrietta and Barry Lubbe of Mertenhof farm and Barend Salomo and all members of the Wupperthal Original Rooibos Cooperative for permission and support in sampling *Cyanella alba*. We thank Sinéad Watchorn and Lia Hemerik for support in the field sampling. We thank Christian Kappel and Mathias Scharmann for suggestions on genome assembly and transcriptome analysis. Some computations were performed using facilities provided by the University of Cape Town's ICTS High Performance Computing team. C.R. held a Harry Crossley Research Fellowship. M.H.L.K. was funded by the Nederlandse Organisatie voor Wetenschappelijk Onderzoek (GSGT.2019.019). This work was supported by a grant from the Human Frontiers Science Program to E.D., N.I., S.C.H.B. and M.L. (grant number RGP0036/2021).

## Author contributions

S.C.H.B., E.D., N.I., and M.L. conceptualized the study. C.R., H.X., M.S., A.L.M., V.W., R.A.I, S.C.H.B., E.D., N.I., and M.L. designed experiments. C.R., H.X., M.S., A.L.M., D.L., M.H.L.K., R.A.I, S.C.H.B., E.D., N.I., and M.L. performed experiments and analysed the data. M.H.L.K. and V.W. provided analysis tools. M.S., S.C.H.B., E.D., N.I., and M.L. wrote the manuscript with input from all authors. S.C.H.B., E.D., N.I., and M.L. acquired funding.

## Funding

## Competing interests

The authors declare no competing interests.
