## [Transparent Peer Review file · Nature Communications]

Spiral phyllotaxis predicts left-right asymmetric growth and style deflection in mirror-image flowers of *Cyanella alba*

Corresponding Author: Professor Michael Lenhard

Version 0:

Reviewer comments:

Reviewer #1

(Remarks to the Author)

This manuscript presents three major findings: (1) The polymorphism in floral handedness (enantiostyly) in *Cyanella alba* does NOT have a genetic basis; instead, it's developmentally plastic and can switch direction within an individual between flowering seasons; (2) The floral handedness seems to be associated with the handedness of the phyllotactic spiral; (3) The direction of style deflection is determined by differential elongation of the two adaxial carpels, which is likely mediated through differential auxin signaling. The manuscript is well written, easy to read, with attention to details. I will focus my comments on the three main findings.

Among these three findings, the first one is the most interesting, and to some degree, unexpected. Although the lack of a genetic basis is a "negative" result in a conventional sense, it's actually more revealing, demonstrating that this kind of prominent phenotypic polymorphism in nature can simply be caused by some random events during development. The thorough genomic and transcriptomic analyses as well as careful field observation of tagged plants provided convincing evidence to support this first conclusion.

On the second finding: I am convinced that the floral handedness is CORRELATED with the handedness of the phyllotactic spiral, but I don't think there is strong evidence suggesting the latter DETERMINING the former. First of all, the correlation between the two is not overwhelmingly strong (correlation coefficient is 0.82). Second, there is no experimental data supporting a causal relationship. Is there a relatively easy way to manipulate the handedness of the phyllotactic spiral (e.g., by mechanical force or NPA treatment)? I understand these plants are in the wild and only flower in certain months, so these experiments may not be feasible. If that's the case, I would suggest to at least tone down the conclusion of this section, pointing out despite the appealing correlation, a causal relationship remains to be tested in future studies.

On the third finding: data supporting the conclusion that "style deflection results from differential elongation of the two adaxial carpels" are solid and the modeling part is a nice addition. The inference that the differential elongation of the two adaxial carpels is mediated through differential auxin signaling makes sense, adding to an increasing body of literature implicating the role of differential auxin signaling in floral organ movement (e.g., sunflower solar tracking, Atamian et al. 2016. *Science* 353: 587-590; delphinium petal movement, Zhang et al., 2024. *Current Biology* 34: 755-768). However, this study suffers from the same limitations as those previous studies on non-model organisms: there is no direct auxin sensor reporting the dynamics of auxin concentration or signaling during organ development. This technical hurdle probably cannot be overcome in a system like *Cyanella alba*. In that case, I would suggest to be more cautious with the wording when discussing auxin dynamics. For example, line 554-556 implies that we "know" auxin concentration is asymmetrically distributed in the floral meristem, and this asymmetry has been maintained during the long process of pistil development. But we don't really know that. In fact, my guess is that there is no such a stable auxin asymmetry during pistil morphogenesis, as auxin signaling is extremely dynamic during organ development.

Taken together, these three findings represent a valuable contribution to our limited understanding of enantiostyly. However, we have to acknowledge that enantiostyly is a relatively rare phenomenon. I like the study because I personally believe in the value of studying peculiar/wacky/uncommon systems. I suspect not everyone feels the same way. Therefore, it might be

useful to spend a few sentences in the INTRODUCTION or/and DISCUSSION to talk about how studies of such "wacky" systems can help inform us with more general principles.

Reviewer #2

(Remarks to the Author)

The authors have combined cellular analysis with data-driven mechanical modelling to elucidate the main drivers behind style deflection in a representative species of mirror-image flowers. It was found that in *Cyanella alba*, the floral handedness in question is not genetically determined; instead, differential elongation of carpels determines the direction of style deflection. This work is original and constitutes a significant advance in the understanding of LR asymmetry in flowers.

I have some reservations about the soundness of the mathematical methods employed in the study, or the clarity with which the methods are explained. My concerns and questions are as follows and will refer exclusively to the supplementary material, since that is where all the mathematical methodologies are found.

In equation S4, the third line gives a nonzero contribution to $T(x)$ if and only if two conditions are met simultaneously: $x > \sqrt{3}l_s/2 + (l_t - l_s)/\sqrt{3}$, and $x < \sqrt{3}l_s/2$. These two conditions cannot possibly both hold as long as $l_t > l_s$. Can the authors double-check this equation?

For equation S10, the authors admit that directly seeking the equilibrium solution is too computationally expensive, and therefore an iterative method was employed to find an approximation. It is important to check the accuracy of such approximations, which the authors seem to have neglected to do. The approximate solution should be put into equation S10 to verify that the resulting right-hand side is sufficiently close to zero. This is the direct way to validate the approximation. Instead of this, the authors have used an indirect validation criterion (equation S15) which seems to be quite arbitrary.

Equation S10 takes the general form of a standard masses-and-springs system, but the terms l_{jm} , as defined by Equation S11c, are clearly not the initial lengths of the springs. Can the authors explain these terms?

To approximate the time-average of $z_j[n](t)$ (equation S14), the authors have taken $T=200$; can the authors justify this choice? Indeed, if the function $z_j[n](t)$ is periodic in t (as the authors have claimed), one should take T to be the period.

If the authors can address these issues without significantly altering the model predictions and hence the main results, then I recommend this paper for publication.

Reviewer #3

(Remarks to the Author)

The paper titled "Style deflection is determined by the handedness 1 of phyllotaxis and auxin-induced 2 differential cell elongation in a species with mirror-image flowers" is an interesting paper. It tells a nice story about the importance of growth, mechanics, and development and how we need to understand these concepts to understand plants, not just their genetics.

The author's article begins by showing that there is no genetic basis to explain the floral left hand and right handedness, which is an apparently random decision between growing seasons. They then correlate the direction of asymmetry with the phyllotaxis direction. The authors then move on to offer an explanation as to how the style bends and find that two adaxial carpels differential elongate. For this, they created a Bead-spring mechanical model and fitted it to the data to show that differential growth and stiffness are required to cause the observed bending. This model is then validated with further experimental data to show this differential growth. The article finishes with an explanation of how this differential growth occurs by demonstrating that there are higher levels of genes related to auxin and other growth-inducing factors in the faster-growing carpel. Following up, they then apply auxin to the slower growing carpel to show that they can get rid of this bending/make it bend the other way.

I recommend this article for publication. Some major corrections need to happen before I can accept it for publication, and some minor points also need to be addressed. These are listed below.

Major points

- In this article, from the modelling and data, you argue that the ovary is driving the bending. I believe all this, but you have not ruled out the style's contribution. So you still need to do the same data analysis of the growth of the style (data collection of differential growth), as that could potentially cause bending as well (without this data, we don't know this information, meaning it could be the style that drives the majority of the bending) and is, therefore, a hole in your argument. The video shows changes in the style curvature. So you need to demonstrate that the style does not contribute to the bending or that the ovary contributes more (which, by just being at the base, will allow more deflection of the tip for the same angle movement)
- Figure 4: Your use of the cell length as an elongation metric must be changed as it is not a valid way to measure growth. Imagine a cell on one side that is of length 1 and another cell on the other also of length 1 and that they are both growing at the same rate to get a total length of 2. If the cell on the other side divides, then both of these cells will have length 1, but the total length will still be the same. Your metric will make it seem like the other side has grown less. It is, therefore, essential that you change this to something like the growth rate of each cell

Minor points

- Line 179: Figure 1D referenced but there is no figure 1D
- Paragraph starting at 174: I think it would be good/strengthen the story to point out that yes, this is not genetically controlled, but it is also not random. If this were a random process, your plant R proportion would be around 0.5.
- Line 225: My above bullet point is especially important when you use the word non-random here; this claim needs backing up, but I think it is enough to comment that a random process would produce a proportion of 0.5
- Line 221: Of these plants that produce both L and R, were the majority of the flowers consistent with phyllotaxis? If so, this could help your argument, and if not, I think it should be declared.
- Line 224: "determines" is too strong a word to use in a scientific article. I agree with the conclusion, but it needs to be softened, as you can't be 100% sure.
- Paragraph beginning at line 227: It would be good to reference Figure 2 somewhere in this text, as that is the data you are referring to
- Line 302: The prediction of compression can be tested to validate your model further. Measure the compressed side while still connected to the tissue, then dissect and separate to see if this tissue is longer on its own.
- Figure 3F: I am not a fan of this figure, as from my understanding of the model, this figure's depiction of it is misleading. The first impression it gives is not helpful to a reader. Here you show 3 springs on the left hand side in series and 2 on the right (I assumed the wiggly lines are springs). Upon first seeing it, it worried me about the model's validity, as springs in series have different properties depending on the number of springs in sequence. Especially with the next figure showing different elongation rates and stiffness. This figure gives the interpretation that you build a bead-spring network depending on the exact outline of the ovary, with the style being at the top bead. After reading the supplementary material, I then knew the structure of the bead network (i.e. a, b, c, alpha, beta, gamma nodes) is always the same, with the style being between alpha and beta. I therefore suggest you change this figure to avoid this misinterpretation. Maybe by clearly showing the labels and how they are moving or showing that the top shortest one is where the style always goes.
- Line 497: Agree that you got that the base is stiff from the model predictions but you have not tested whether this is essential for the bending. This is, therefore, a strong claim to say that this development process is coupled with a strong base. You could run simulations with different base stiffness as controls to show how important the stiff base is.
- Supplementary material page 6: In the first paragraph, do you mean to refer to figure 3, not figure 4?
- Supplementary material page 8: Table s8, no errors are mentioned or shown for the different fits, so the reader has no idea how accurate your fits are.

Version 1:

Reviewer comments:

Reviewer #1

(Remarks to the Author)

The authors have addressed all of my comments. I particularly enjoy the new ending of the DISCUSSION.

Reviewer #2

(Remarks to the Author)

The authors have adequately addressed all my comments relating to the mathematical methodologies. I therefore recommend the manuscript for publication.

Reviewer #3

(Remarks to the Author)

The authors have addressed all the issues raised in the previous round of revisions to my satisfaction. I therefore recommend this article for publication.

RESPONSE TO REVIEWER COMMENTS

We are grateful to all three reviewers for their positive comments and their constructive feedback. We have addressed all comments and are convinced that this has substantially improved the manuscript.

Reviewer #1 (Remarks to the Author):

This manuscript presents three major findings: (1) The polymorphism in floral handedness (enantiostyly) in *Cyanella alba* does NOT have a genetic basis; instead, it's developmentally plastic and can switch direction within an individual between flowering seasons; (2) The floral handedness seems to be associated with the handedness of the phyllotactic spiral; (3) The direction of style deflection is determined by differential elongation of the two adaxial carpels, which is likely mediated through differential auxin signaling. The manuscript is well written, easy to read, with attention to details. I will focus my comments on the three main findings.

Among these three findings, the first one is the most interesting, and to some degree, unexpected. Although the lack of a genetic basis is a "negative" result in a conventional sense, it's actually more revealing, demonstrating that this kind of prominent phenotypic polymorphism in nature can simply be caused by some random events during development. The thorough genomic and transcriptomic analyses as well as careful field observation of tagged plants provided convincing evidence to support this first conclusion.

>>Thank you for these positive comments.

On the second finding: I am convinced that the floral handedness is CORRELATED with the handedness of the phyllotactic spiral, but I don't think there is strong evidence suggesting the latter DETERMINING the former. First of all, the correlation between the two is not overwhelmingly strong (correlation coefficient is 0.82). Second, there is no experimental data supporting a causal relationship. Is there a relatively easy way to manipulate the handedness of the phyllotactic spiral (e.g., by mechanical force or NPA treatment)? I understand these plants are in the wild and only flower in certain months, so these experiments may not be feasible. If that's the case, I would suggest to at least tone down the conclusion of this section, pointing out despite the appealing correlation, a causal relationship remains to be tested in future studies.

>>We agree that we have not demonstrated causality. As the reviewer states, experiments to manipulate the direction of phyllotaxis are unfeasible, because the plants are geophytes and the shoot meristems for next season's inflorescences seem to develop underground, making them very difficult to access. Therefore, following the reviewer's suggestion, we have toned down the conclusion of this section accordingly. We also no longer mention 'determine' in the title, which we needed to shorten to conform to the journal instructions.

On the third finding: data supporting the conclusion that "style deflection results from differential elongation of the two adaxial carpels" are solid and the modeling part is a nice addition.

>>We thank the reviewer for the positive comment and the recognition of the modelling component. We believe our model has intrinsic value as an analytical tool, because it both supports the observed experimental findings and provides predictions (e.g., identifying the location of the most intense cell growth within a carpel). It also offers a framework for future investigations into similar phenomena, which may have significance beyond this study.

The inference that the differential elongation of the two adaxial carpels is mediated through differential auxin signaling makes sense, adding to an increasing body of literature implicating the role of differential auxin signaling in floral organ movement (e.g., sunflower solar tracking, Atamian et al. 2016. *Science* 353: 587-590; delphinium petal movement, Zhang et al., 2024. *Current Biology* 34: 755–768). However, this study suffers from the same limitations as those previous studies on non-model organisms: there is no direct auxin sensor reporting the dynamics of auxin concentration or signaling during organ development. This technical hurdle probably cannot be overcome in a system like *Cyanella alba*. In that case, I would suggest to be more cautious with the wording when discussing auxin dynamics. For example, line 554-556 implies that we "know" auxin concentration is asymmetrically distributed in the floral meristem, and this asymmetry has been maintained during the long process of pistil development. But we don't really know that. In fact, my guess is that there is no such a stable auxin asymmetry during pistil morphogenesis, as auxin signaling is extremely dynamic during organ development.

>>We agree that the absence of an auxin sensor to show actual auxin distribution is a limitation, but also that it cannot be solved at the moment. As suggested, we have mentioned this limitation in the discussion (lines 546-547) and also changed the wording about the link between a presumed auxin asymmetry in the primordium and the later elaboration of style deflection in lines 571/572 and 575.

Taken together, these three findings represent a valuable contribution to our limited understanding of enantiostyly. However, we have to acknowledge that enantiostyly is a relatively rare phenomenon. I like the study because I personally believe in the value of studying peculiar/wacky/uncommon systems. I suspect not everyone feels the same way. Therefore, it might be useful to spend a few sentences in the INTRODUCTION or/and DISCUSSION to talk about how studies of such "wacky" systems can help inform us with more general principles.

>>Thank you for this suggestion. We have rephrased and expanded the last section of the discussion (lines 623 onward) to point out the value of studying more unusual systems for obtaining novel perspectives on general principles of development, also drawing on your insightful comment above.

Reviewer #2 (Remarks to the Author):

The authors have combined cellular analysis with data-driven mechanical modelling to elucidate the main drivers behind style deflection in a representative species of mirror-image flowers. It was found that in *Cyanella alba*, the floral handedness in question is not genetically determined; instead, differential elongation of carpels determines the direction of style deflection. This work is original and constitutes a significant advance in the understanding of LR asymmetry in flowers.

I have some reservations about the soundness of the mathematical methods employed in the study, or the clarity with which the methods are explained. My concerns and questions are as follows and will refer exclusively to the supplementary material, since that is where all the mathematical methodologies are found.

>> We thank the reviewer for the positive comment. Please see below our responses related to the mathematical methodology.

In equation S4, the third line gives a nonzero contribution to $T(x)$ if and only if two conditions are met simultaneously: $x > \sqrt{3}l_s/2 + (l_t - l_s)/\sqrt{3}$, and $x < \sqrt{3}l_s/2$. These two conditions cannot possibly both hold as long as $l_t > l_s$. Can the authors double-check this equation?

>>The second Heaviside theta in Equation S4 should read $\sqrt{3} l_t / 2 - x$. We fixed it. Thank you for spotting this typo.

For equation S10, the authors admit that directly seeking the equilibrium solution is too computationally expensive, and therefore an iterative method was employed to find an approximation. It is important to check the accuracy of such approximations, which the authors seem to have neglected to do. The approximate solution should be put into equation S10 to verify that the resulting right-hand side is sufficiently close to zero. This is the direct way to validate the approximation. Instead of this, the authors have used an indirect validation criterion (equation S15) which seems to be quite arbitrary.

>> We agree with the reviewer that checking the accuracy of our chosen approximation is important. One of the ways to do so is, indeed, verifying that the resulting r.h.s. is sufficiently close to zero. Even though we did not include that in the text, we have already performed such an analysis (see leftmost panel of Fig. S6 for an example). We have now added further panels to Figure S6, one for each image analyzed, where we show that at the end of the iterative process, our solution has minimal oscillations, i.e., the approximated solution is sufficiently close to 0. Furthermore, we added text to the relevant section in the SI.

Equation S10 takes the general form of a standard masses-and-springs system, but the terms l_{jm} , as defined by Equation S11c, are clearly not the initial lengths of the springs. Can the authors explain these terms?

>>We added an explanatory sentence after Equation S10 and a new equation (S11) where we give the expression for the initial length of the springs.

To approximate the time-average of $z_j[n](t)$ (equation S14), the authors have taken $T=200$;

can the authors justify this choice? Indeed, if the function $z_j[n](t)$ is periodic in t (as the authors have claimed), one should take T to be the period.

>>Although we detect an oscillatory solution, our results do not establish that this solution is periodic. Consequently, calculating the exact time average requires extending T to infinity. In this context, the choice of $T=200s$ is practical: it is sufficiently long to provide a reliable approximation of the time average while remaining short enough to allow for the numerical computation of the solution of the system of ODE within a reasonable timeframe. For example, in the case of the image labeled R2, $z_a[n](t)$ averaged over 20s is $z_a=3.57643$, over 200s is $z_a=3.57603$, and over 2000s is $z_a=3.57600$. We expanded our explanation for our choice of $T=200s$. We do not believe it is necessary to add a full analysis of the quality of this approximation to the supplementary information. However, if the reviewer considers it essential, we are happy to include more a detailed proof in the supplementary material.

If the authors can address these issues without significantly altering the model predictions and hence the main results, then I recommend this paper for publication.

Reviewer #3 (Remarks to the Author):

The paper titled "Style deflection is determined by the handedness 1 of phyllotaxis and auxin-induced 2 differential cell elongation in a species with mirror-image flowers" is an interesting paper. It tells a nice story about the importance of growth, mechanics, and development and how we need to understand these concepts to understand plants, not just their genetics.

The author's article begins by showing that there is no genetic basis to explain the floral left hand and right handedness, which is an apparently random decision between growing seasons. They then correlate the direction of asymmetry with the phyllotaxis direction. The authors then move on to offer an explanation as to how the style bends and find that two adaxial carpels differential elongate. For this, they created a Bead-spring mechanical model and fitted it to the data to show that differential growth and stiffness are required to cause the observed bending. This model is then validated with further experimental data to show this differential growth. The article finishes with an explanation of how this differential growth occurs by demonstrating that there are higher levels of genes related to auxin and other growth-inducing factors in the faster-growing carpel. Following up, they then apply auxin to the slower growing carpel to show that they can get rid of this bending/make it bend the other way.

I recommend this article for publication. Some major corrections need to happen before I can accept it for publication, and some minor points also need to be addressed. These are listed below.

Major points

- In this article, from the modelling and data, you argue that the ovary is driving the bending. I believe all this, but you have not ruled out the style's contribution. So you still need to do the same data analysis of the growth of the style (data collection of differential growth), as that could potentially cause bending as well (without this data, we don't know this information, meaning it could be the style that drives the majority of the bending) and is, therefore, a hole in your argument. The video shows changes in the style curvature. So you need to demonstrate that the style does not contribute to the bending or that the ovary contributes more (which, by just being at the base, will allow more deflection of the tip for the same angle movement)

>>Thank you for pointing this out. We agree that in our earlier version we did not rule out a contribution of differential carpel elongation in the style part to style deflection from the midline. We have now addressed this issue as follows (lines 408-421):

“Our above model considers differential carpel elongation in the ovary as the main driver for style deflection. It is conceivable that such differential elongation also occurs within the style itself and contributes to its deflection. Addressing such a contribution would require measuring the length of the style along the three carpels; however, both the measurements and their interpretation would be confounded by the upward bend of the style at its tip (Figure 1A). Differential elongation of the style part of the three carpels (ignoring the upward bend at the tip) should result not only in deflection of the style away from the midline, but also in its curvature. To assess this possibility, we defined three points (style/ovary transition, beginning of upward bend at the style tip, and midpoint between these) and measured the angle between

the two lines connecting these points (Figure S9). At 175° and 176° these angles were very close to the 180° of a perfectly straight style in both left- and right-handed flowers (Figure S9). Thus, although we cannot exclude a contribution of differential carpel elongation within the style to overall style deflection, such a contribution - if any - appears to be small relative to the role played by differential growth in the ovary.”

In other words, we believe that it would not be possible to derive meaningful measures of the lengths of the three carpels in the style, because the upward bending at the tip is necessarily associated with a differential relative elongation of the three carpels in this region, which will confound the measure. At the same time, measuring the length from the ovary/style transition to just before the upward bend would require setting an arbitrary end point for these measurements. Therefore, it seems to us that the measures of style curvature (or rather straightness) that we have now included provide the most feasible, albeit indirect way of addressing the reviewer’s concern.

- Figure 4: Your use of the cell length as an elongation metric must be changed as it is not a valid way to measure growth. Imagine a cell on one side that is of length 1 and another cell on the other also of length 1 and that they are both growing at the same rate to get a total length of 2. If the cell on the other side divides, then both of these cells will have length 1, but the total length will still be the same. Your metric will make it seem like the other side has grown less. It is, therefore, essential that you change this to something like the growth rate of each cell

>>We agree that cell length would not be a suitable measure on its own, and apologize, if this was not clear. The differential elongation of the three carpels shown in Figure 4B was measured based on the lengths of the three carpel midveins. We performed the cell length measurements to ask whether this differential carpel elongation could be fully explained by differential cell elongation or whether cell division differences also play a role. Here we found that the difference in cell lengths between carpels 1 and 2 can fully explain the difference in their overall lengths. We have now rephrased the beginning of the relevant section to make it clear that the differential carpel elongation was shown by measuring carpel midvein lengths (lines 386-387).

Minor points

- Line 179: Figure 1D referenced but there is no figure 1D

>>Thanks, corrected.

- Paragraph starting at 174: I think it would be good/strengthen the story to point out that yes, this is not genetically controlled, but it is also not random. If this were a random process, your plant R proportion would be around 0.5.

>>Thanks, we have added this point to this paragraph (lines 174-177).

- Line 225: My above bullet point is especially important when you use the word non-random here; this claim needs backing up, but I think it is enough to comment that a random process would produce a proportion of 0.5

>>Thanks, we have added the argument for why this is not a random process to the section above (lines 174-177).

- Line 221: Of these plants that produce both L and R, were the majority of the flowers consistent with phyllotaxis? If so, this could help your argument, and if not, I think it should be declared.

>>Yes, 75% of the ‘mixed’ plants that produced more than two flowers had a majority of flowers that was consistent with the prediction based on phyllotaxis. We have added this information (lines 221-223).

- Line 224: "determines" is too strong a word to use in a scientific article. I agree with the conclusion, but it needs to be softened, as you can't be 100% sure.

>>Thanks, we have altered this to ‘predicts’ throughout.

- Paragraph beginning at line 227: It would be good to reference Figure 2 somewhere in this text, as that is the data you are referring to

>>Thanks, we have added a reference to Figure 2.

- Line 302: The prediction of compression can be tested to validate your model further. Measure the compressed side while still connected to the tissue, then dissect and separate to see if this tissue is longer on its own.

>>Thanks for the suggestion. We agree that this would be an informative experiment. However, it would require waiting almost 9 months until the plants flower again, and we are not convinced the additional insight justifies such a long delay.

- Figure 3F: I am not a fan of this figure, as from my understanding of the model, this figure's depiction of it is misleading. The first impression it gives is not helpful to a reader. Here you show 3 springs on the left hand side in series and 2 on the right (I assumed the wiggly lines are springs). Upon first seeing it, it worried me about the model's validity, as springs in series have different properties depending on the number of springs in sequence. Especially with the next figure showing different elongation rates and stiffness. This figure gives the interpretation that you build a bead-spring network depending on the exact outline of the ovary, with the style being at the top bead. After reading the supplementary material, I then knew the structure of the bead network (i.e. a, b, c, alpha, beta, gamma nodes) is always the same, with the style being between alpha and beta. I therefore suggest you change this figure to avoid this misinterpretation. Maybe by clearly showing the labels and how they are moving or showing that the top shortest one is where the style always goes.

>>We apologize for this lack of clarity. We have now made the lines indicating the base of the style thicker in Figures 3C to 3F and indicated this in the legend (lines 320-321). Additionally, we have indicated a few more springs in figure 3F, the internal ones of those in grey, to visually indicate which parameters were optimized. We have updated the corresponding description in the caption of 3F (lines 325-327).

- Line 497: Agree that you got that the base is stiff from the model predictions but you have not tested whether this is essential for the bending. This is, therefore, a strong claim to say that this development process is coupled with a strong base. You could run simulations with different base stiffness as controls to show how important the stiff base is.

>> We agree with the reviewer that, based on our model predictions, our claim that strong stiffness of the style is essential for bending may have been somewhat overstated. To address this, we have conducted the suggested analysis. Specifically, for each ovary considered in the model, we compared the bending angle at the base of the style using the optimised parameters to the angle obtained when the stiffness at the base of the style was lower. Our new results show that, indeed, with lower stiffness, stylar bending is exaggerated. Therefore, we conclude that high stiffness at the base of the style is a necessary factor for reproducing the observed bending angle.

This analysis has been included in the supplementary material, accompanied by a new Table S9. Additionally, we have revised the manuscript to reflect these findings. In the Discussion section, we have modified the relevant sentence (lines 516-517) to read: “Developmentally, style deflection is caused by differential cell elongation between the two adaxial carpels, coupled with a very stiff base to the ovary **as inferred from our biophysical model.**” In the Results section, we have added a new sentence (lines 301-303): “Indeed, running our model with lower stiffness values generates stylar bending that far exceeds the recorded and realistic deflection angles (Table S9).”

- Supplementary material page 6: In the first paragraph, do you mean to refer to figure 3, not figure 4?

>>Thanks for pointing this out. The reference should indeed be to Figure 3B. We have changed this.

- Supplementary material page 8: Table s8, no errors are mentioned or shown for the different fits, so the reader has no idea how accurate your fits are.

>> Given that, in our geometrical optimisation, all edges of the polyhedra fitting the ovary shapes were free parameters, our six optimisation processes were conducted with a very high level of accuracy - the worst fit had a mean relative squared error (MRSE) of 0.00013. We have added this remark before introducing Table S4 (sorry, there was an error in the table numbers in the former version of the SI). Similarly, we have supplemented Table S7 with an extra column showing the MRSE of the target function of the optimisation problem for the mechanistic model compared to the data-driven model.